# Virus-modified paraspeckle-like condensates are hubs for viral RNA processing and their formation drives genomic instability

Katherine L. Harper [1,6], Elena M. Harrington[1,6], Connor Hayward [1], Chinedu A. Anene [2,3], Wiyada Wongwiwat[4], Robert E. White [4] & Adrian Whitehouse [1,5] ✉

The nucleus is a highly organised yet dynamic environment containing distinct membraneless nuclear bodies. This spatial separation enables a subset of components to be concentrated within biomolecular condensates, allowing efficient and discrete processes to occur which regulate cellular function. One such nuclear body, paraspeckles, are comprised of multiple paraspeckle proteins (PSPs) built around the architectural RNA, *NEAT1_2*. Paraspeckle function is yet to be fully elucidated but has been implicated in a variety of developmental and disease scenarios. We demonstrate that Kaposi's sarcoma-associated herpesvirus (KSHV) drives formation of structurally distinct paraspeckles with a dramatically increased size and altered protein composition that are required for productive lytic replication. We highlight these virus-modified paraspeckles form adjacent to virus replication centres, potentially functioning as RNA processing hubs for viral transcripts during infection. Notably, we reveal that PSP sequestration into virus-modified paraspeckles result in increased genome instability during both KSHV and Epstein Barr virus (EBV) infection, implicating their formation in virus-mediated tumourigenesis.

The nucleus is an extremely crowded but highly organised environment, containing multiple distinct nuclear bodies which each fulfil specific roles and functions. These membraneless organelles are formed through liquid-liquid phase separation (LLPS), allowing spatial separation and concentration of specific components enabling efficient discrete processes to regulate gene expression[1]. Paraspeckles, discovered relatively recently in 2002, are one such nuclear body, located in the interchromatin space[2]. Canonical paraspeckles form through core paraspeckle protein (PSP) interactions with the architectural lncRNA *NEAT1_2* producing a unique structure comprising of an outer shell and inner core[3]. Biogenesis occurs co-transcriptionally, with rapid binding of the core PSPs SFPQ and NONO aiding in *NEAT1_2*

stability. Oligomerisation of these PSPs along the ~23 kb transcript leads to the formation of a stable ribonucleoprotein particle (RNP)[4,5]. LLPS occurs as multiple RNPs link together through the protein FUS, leading to the formation of mature paraspeckles[6]. There are estimated to be around 100 proteins that dynamically associate with paraspeckles, categorised as either essential, important or dispensable for paraspeckle formation[5,7]. The majority of mammalian tissues and cells contain paraspeckles, however depending on cell type paraspeckles can vary in number, typically ranging between 5 and 20 foci per nucleus[8]. The paraspeckle function has yet to be fully determined but they have been implicated in a range of cellular processes, such as the regulation of gene expression by sequestration of RNA and proteins

[1]School of Molecular and Cellular Biology and Astbury Centre for Structural Molecular Biology, University of Leeds, Leeds LS2 9JT, UK. [2]Centre for Biomedical Science Research, School of Health, Leeds Beckett University, Leeds LS1 3HE, UK. [3]Centre for Cancer Genomics and Computation Biology, Barts Cancer Institute, Queen Mary University of London, London EC1M 6AU, UK. [4]Department of Infectious Disease, Imperial College London, South Kensington Campus, London SW7 2AZ, UK. [5]Department of Biochemistry & Microbiology, Rhodes University, Grahamstown 6140, South Africa. [6]These authors contributed equally: Katherine L. Harper, Elena M. Harrington. ✉e-mail: a.whitehouse@leeds.ac.uk

which, when dysregulated, lead to various disease scenarios, such as neurodegeneration and cancer[9,10]. This role complements their dynamic structure, with their formation, function and dissolution orchestrating a fine-tuned cellular response to specific stimuli[11–14].

Kaposi's sarcoma-associated herpesvirus (KSHV) is a large dsDNA gammaherpesvirus associated with the development of Kaposi's sarcoma (KS), a highly vascular tumour of endothelial lymphatic origin, and several other lymphoproliferative diseases including primary effusion lymphoma (PEL) and some forms of multicentric Castleman's disease (MCD)[15–19]. KSHV exhibits a biphasic life cycle consisting of latent persistence and lytic replication. Latency is established in B cells and in the tumour setting, where viral gene expression is limited to the latency-associated nuclear antigen (LANA), viral FLICE inhibitory protein, viral cyclin, kaposins and several virally-encoded miRNAs[20–22]. Upon reactivation, KSHV enters the lytic replication phase, leading to the highly orchestrated expression of more than 80 viral proteins that are sufficient to produce infectious virions[23,24]. In KS lesions, most infected cells harbour the virus in a latent state. However, a small proportion of cells undergo lytic or abortive lytic replication that leads to the secretion of angiogenic, inflammatory and proliferative factors that act in a paracrine manner on latently infected cells to enhance tumorigenesis[25]. Lytic replication also sustains KSHV episomes in latently-infected cells that would otherwise be lost during cell division[26,27] and manipulation of cellular RNA processing pathways required for efficient lytic replication leads to genomic instability[28]. Therefore, both the latent and lytic replication phases are implicated in KSHV-mediated tumourigenicity, and the ability to inhibit the lytic replication phase represents a therapeutic intervention strategy for the treatment of KSHV-associated diseases[29,30].

During the early stages of herpesvirus lytic replication, the nuclear architecture of the host cell undergoes a striking re-organisation to facilitate viral replication. This is driven by the formation of virus-induced nuclear structures, termed virus replication and transcription compartments (vRTCs), which support viral transcription, DNA replication and packaging and capsid assembly[31,32]. As lytic replication proceeds, small vRTCs coalesce into a singular large globular structure that ultimately fill most of the nuclear space compressing and marginalising the cellular chromatin to the nuclear periphery. How this drastic virus-induced remodelling of the nuclear architecture affects the localisation and function of nuclear bodies to regulate gene expression is yet to be fully elucidated. In particular, how paraspeckles are impacted is of prominent interest as their dynamic formation is emerging as a global sensor of cellular stress[33]. Notably, multiple stressors including mitochondrial stress, hypoxia and heat shock can induce changes in paraspeckle abundance, which is thought to have a cytoprotective role in response to these stressors[33]. This is particularly intriguing as understanding the intricate interplay between paraspeckle components and viruses is still in its infancy[34]. Although, elucidating paraspeckle function and their association with disease is complicated by the dynamic nature of their composition, and the fact PSPs are multifunctional and not exclusively confined to paraspeckles.

This study demonstrates that KSHV drives the formation of novel, non-canonical paraspeckles during lytic replication, which we term virus-modified paraspeckles (v-mPS). Specifically, we show for the first time that v-mPS are altered in their localisation, aligning adjacent to vRTCs, and are approximately 10 times larger than the size of standard paraspeckles. Further evidence supports that these structures are modified and distinct from canonical paraspeckles, confirmed by their unique protein composition including the association of the multi-functional KSHV-encoded ORF11 protein, which appears to be essential for v-mPS formation during infection. Notably, we also show that disruption of v-mPS formation leads to a reduction in virus replication and infectious virion production. We further show that viral transcripts are associated with multiple paraspeckle components within v-mPS,

which suggest they act as hubs for the processing of viral RNAs required for efficient KSHV lytic replication. Surprisingly, enlarged v-mPS were not observed during HSV-1 or HCMV infection, whereas EBV lytic replication resulted in similar enlarged v-mPS. Importantly, aligned with the unique gamma herpesvirus specificity, a correlation is observed between the formation of v-mPS and increased genomic instability, which may indirectly play a role in gamma herpesvirus-mediated oncogenesis[29]. Taken together, this study has identified novel v-mPS forming during KSHV infection which enhance lytic replication and are implicated in virus-mediated genomic instability.

## Results

### KSHV induces the formation of novel puncta during lytic replication

The host cell nuclear architecture undergoes dramatic remodelling to facilitate KSHV lytic replication, due to the formation of vRTCs which support viral transcription, DNA synthesis and capsid assembly[31,35]. We therefore aimed to investigate how this reorganisation affects nuclear bodies, with particular focus on the impact of virus-induced remodelling of paraspeckles, a nuclear body whose formation is emerging as a global sensor of cellular stress[11,33]. Interestingly, other cell stress sensors such as cytoplasmic stress granules are disrupted during KSHV lytic replication[36]. Therefore, to examine whether KSHV-mediated remodelling of the nucleus induces changes in paraspeckles, TREx-BCBL1-RTA cells, a KSHV-latently infected B-lymphocyte cell line containing a Myc-tagged version of the viral RTA under the control of a doxycycline-inducible promoter were utilised, allowing efficient induction of the lytic cascade with addition of doxycycline. Cells remained latent or reactivated over a time course prior to immunostaining with antibodies against SFPQ and RNA pol II, markers of paraspeckles and vRTCs, respectively. vRTC development was observed from 8 h post-lytic reactivation and reached full maturity by 24 h, indicated by distinct restructuring of RNA pol II into concentrated areas between the cellular chromatin (Fig. 1A). Surprisingly, a striking change was observed upon staining of the paraspeckle marker, SFPQ. Notably, in KSHV-latently infected TREx-BCBL1-RTA cells, no paraspeckles were observed and SFPQ remained diffuse throughout the nucleus. Whether paraspeckles exist in non-KSHV infected B cells is currently unknown, however paraspeckle formation in response to stress has been reported in many but not all cultured cells, with large amounts of cell type variation[37,38]. In contrast, during the early stages of KSHV lytic replication, SFPQ coalesced into distinct large puncta (Fig. 1A). Whilst these puncta were spatially distinct to the vRTCs, these SFPQ puncta localised adjacent and around them, suggesting they may have a complementary function. Additionally, it is known that paraspeckles often reside close to nuclear speckles, potentially allowing RNA trafficking between these nuclear bodies[2]. Therefore, utilising SRSF2 as a nuclear speckle marker, immunofluorescence was performed which confirmed these SFPQ puncta were distinct from nuclear speckles (Fig. S1A). This suggests that these structures may be KSHV-specific as some paraspeckle components have previously been shown to co-localise with the nuclear speckle protein SRSF2 in HSV-1 infection[8,39]. To investigate whether these puncta were specifically induced by KSHV lytic replication, latently-infected TREx-BCBL1-RTA cells were exposed to a number of stress treatments such as serum starving and osmotic stress, with several of these stress treatments known to induce canonical paraspeckle formation[11,40]. Results confirmed SFPQ puncta formation was a virus-specific response, with no puncta observed in latently-infected stress-treated cells (Fig. S1B). This was further corroborated by formation of SFPQ puncta during lytic replication in an additional KSHV-infected cell line, HEK-293T-rKSHV.219s (Fig. S1C), with no identifiable puncta observed in latent cells or uninfected HEK-293Ts (Fig. S1D). Together these results suggest that KSHV induces the formation of SFPQ containing puncta during the early stages of its lytic replication cycle.

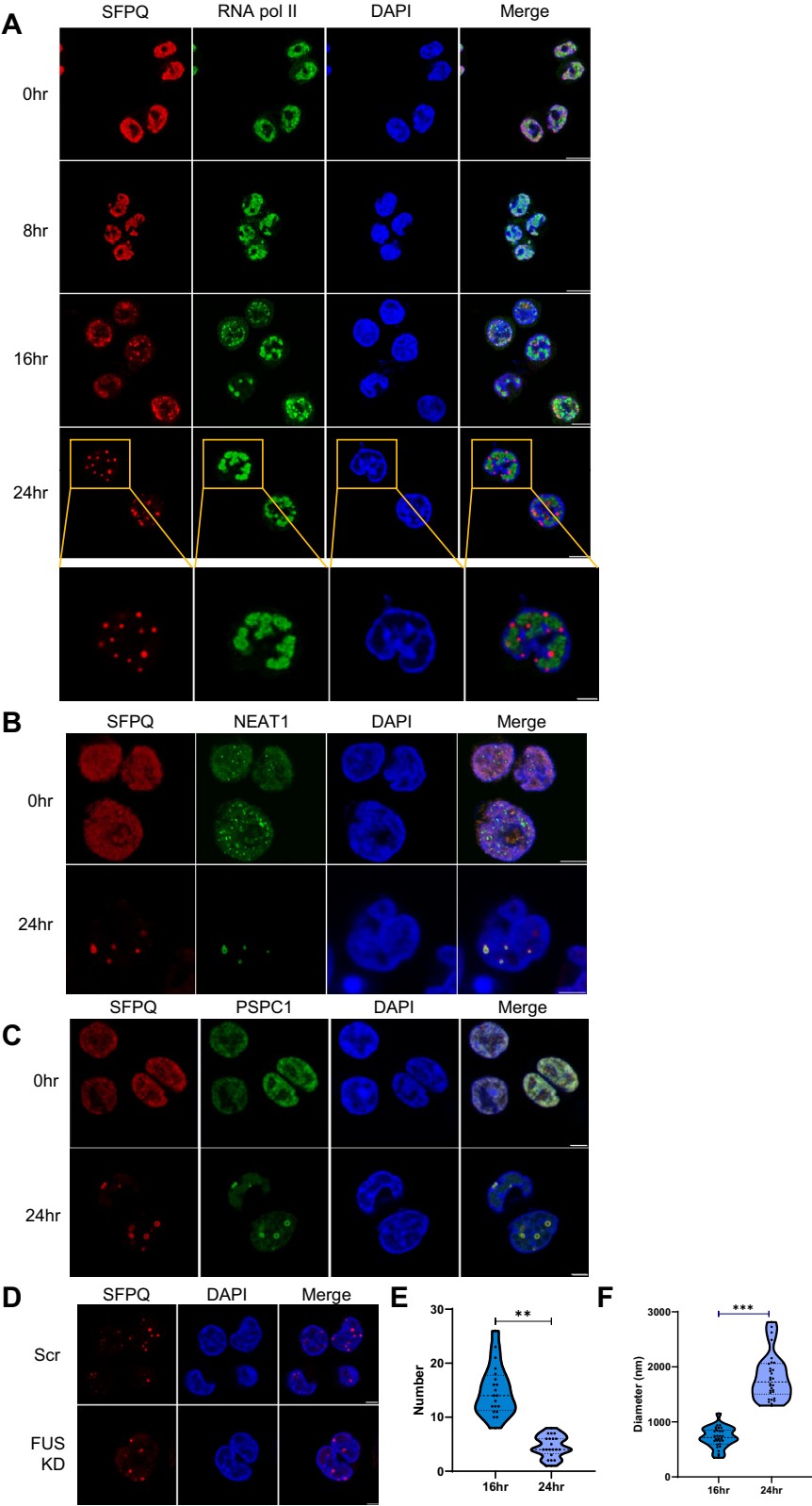

**Fig. 1 | SFPQ forms puncta during KSHV lytic replication. A** IF of TREx cells at 0, 8, 16 and 24 h post-reactivation stained against SFPQ (red), RNA pol II (green) and DAPI (blue). **B**, **C**. IF of TREx-BCBL1-RTA cells at 0 and 24 h stained for SFPQ (red), DAPI (blue) and *NEAT1* (green) (**B**) or SFPQ (red), DAPI (blue) and PSPC1 (green) (**C**). **D** IF of scr and FUS KD TREx at 24 h with staining for SFPQ (red), DAPI (blue). **E** Average number of puncta per z-frame puncta in TREx-BCBL1-RTA cells at 16 and 24 h post-lytic KSHV induction. 20 individual cells were analysed via Zen Blue and FIJI software for each time point. *P* value is <0.0001. **F** Average diameter of puncta in TREx-BCBL1-RTA cells at 16 h or 24 h post-lytic-KSHV induction. 20 individual cells were measured through Zen blue software for each time point. *P* value is <0.0001. Scale bars represent 5 μm except in upper panel (**A**) where scale bars represent 10 μm. Source data are provided as a source data file. Unpaired two-tailed Student's *t* test was used. *P < 0.05, **P < 0.01 and ***P < 0.001.

## KSHV-induced SFPQ-containing puncta are paraspeckle-like structures

SFPQ is a multi-functional RNA binding protein (RBP) with numerous roles in cell regulation, including transcriptional repression, alternative splicing and DNA damage[41,42]. Critically, SFPQ is a core paraspeckle-specific protein (PSP)[43]. Therefore, to determine whether these novel KSHV-induced puncta only contained aggregated SFPQ protein or were modified paraspeckles, RNA-FISH and immunofluorescence studies were performed to determine whether other paraspeckle components were also contained in these novel puncta. Canonical paraspeckles have a sub-structure comprising core and shell zones, containing different PSPs built around the architectural RNA, NEAT1_2 (later referred to as just NEAT1). RNA FISH confirmed the presence and co-localisation of NEAT1 in these novel puncta (Fig. 1B). Notably, there was a lack of co-localisation of SFPQ and NEAT1 within latent TREx-BCBL1-RTA cells (0 h), suggesting this is a virus-specific effect. NEAT1 staining was also performed in HeLa cells, a widely used cell model for paraspeckle research, known to contain multiple paraspeckle foci. In contrast to latent TREx-BCBL1-RTA cells, SFPQ and NEAT1 co-localised in distinct puncta, likely paraspeckles. Notably, these puncta were far smaller than the virus-induced puncta previously observed (Fig. S2A) and unlike reactivated TREx cells, the vast majority of SFPQ was not sequestered into the structures, remaining in the nucleoplasm.

Immunostaining with antibodies against additional characterised PSPs also confirmed PSPC1 and NONO co-localise to the puncta during KSHV lytic replication (Fig. 1C, Fig. S2B). This was further supported by co-immunoprecipitation studies showing that SFPQ interacts with NONO and PSPC1 during both latency and lytic replication, due to the natural affinity of SFPQ to heterodimerise with these proteins (Fig. S2C)[44]. Surprisingly however, certain core paraspeckle proteins were omitted from the virus-induced paraspeckles. FUS, a core PSP, essential for mature paraspeckle formation is absent from the virus-induced paraspeckles, shown using immunofluorescence and Zen analysis instead partly localising to vRTCs (Fig. S2D). This re-localisation was confirmed by co-immunoprecipitation assays which show a decrease in the association of SFPQ and FUS during KSHV lytic replication compared to latent cells (Fig. S2E). Furthermore, to examine whether FUS was required for formation of these novel virus-induced puncta, stable TREx-BCBL1-RTA cells were produced depleting FUS using targeted shRNAs which resulted in a ~ 95% reduction in FUS protein levels (Fig. S2F, G). Results clearly show that paraspeckle-like structures were still capable of forming in the absence of FUS upon reactivation (Fig. 1D, Fig. S2H), in contrast, FUS depletion in HeLa cells, resulted in the failure of NEAT1 and SFPQ to co-localise into canonical paraspeckles (Fig. S2I–K).

This analysis suggests there are distinct differences in the formation and structure of the virus-induced paraspeckles compared to canonical paraspeckles. Therefore, to further investigate the formation of KSHV-induced paraspeckle-like structures, TREx-BCBL1-RTA cells were visualised over a lytic replication time course post-reactivation utilising SFPQ and NEAT1 as markers for the puncta and ORF57 as a marker for virus replication (Fig. S2I–M). Paraspeckle-like puncta formation is observed from as early as 8 h post-reactivation, with the majority forming between 16 and 24 h. During this crucial period, as puncta reduced in number, from an average of 15 to 5 per central Z frame slice (Fig. 1E) they concurrently grew in size, significantly increasing from an average of ~700 nm to ~1800 nm, with the largest up to 3000 nm in diameter, which is 10x larger than canonical paraspeckles (Fig. 1F)[8]. This was further supported by high-resolution Airy-Scan microscopy (Fig. 2A), which showed that SFPQ formed a distinct ring structure.

To investigate the interface between the puncta and vRTCs during viral replication in more detail, STED microscopy was employed staining the respective structures for SFPQ and RNA pol II. Whole cell volume imaging and surface rendering of STED signal for both proteins supported previous observations using standard confocal microscopy (Fig. 2B, C). Optically sectioning nuclei stained for SFPQ and RNA pol II showed a complex 3D arrangement of the virus-induced paraspeckles and vRTCs. Paraspeckle-like structures were observed in close proximity to the exterior surface of vRTCs across multiple sites (Supplementary Movies 1-2). Canonical paraspeckle biogenesis occurs co-transcriptionally with core paraspeckle proteins binding to and stabilising NEAT1, as such NEAT1 levels directly correlate with paraspeckle size[7]. Due to the dramatically increased size of the KSHV-induced paraspeckle-like puncta, it was hypothesised that NEAT1 levels would increase during lytic replication. However, surprisingly, qPCR analysis comparing NEAT1 levels in latent and reactivated TREx-BCBL1-RTA cells, showed no increase in NEAT1 expression (Fig. 2D). Furthermore, qPCR and immunoblot analysis of SFPQ and NONO also showed no significant increase in mRNA (Fig. 2D) or protein levels during lytic replication (Fig. 2E, F). Together these data suggest the puncta are non-canonical paraspeckles, therefore we now refer to them as novel virus-modified paraspeckles (v-mPS).

## v-mPS are dynamic condensates

The dynamic nature of v-mPS formation suggested they may form through LLPS, a property typically associated with condensates. As such, fluorescent recovery after photobleaching (FRAP) was performed using a TREx-BCBL1-RTA cell line over-expressing GFP-SFPQ. Puncta formed at both 16 and 24 h post-lytic replication were exposed to photobleaching to probe their potential dynamicity. Significant fluorescent recovery was observed for puncta formed 16 h post-reactivation, whereas puncta formed after 24 h of lytic replication showed little recovery (Fig. 2G, Fig. S2N). This is suggestive of a shift in the material state of the v-mPS from liquid-like to gel-like, implicating condensate maturation. Further studies utilised the aliphatic alcohol 1,6-hexanediol (1,6HD), which is capable of dissolving liquid-like states, in contrast, gel-like states are more resistant[45]. Similar to the FRAP analysis, v-mPS formed 16 h post-KSHV-lytic reactivation were susceptible to dissolving, whereas v-mPS formed later at 24 h were resistant to 1,6HD treatment, with 17.8% and 91.2% of v-mPS being resistant respectively at each time point (Fig. S2N). Allowing a 30 min recovery period post 1,6HD treatment led to the reforming of v-mPS at 16 h post-lytic induction, confirming their dynamic liquid-like structure during the early stages of KSHV lytic replication (Fig. 2H). Conversely, the recovery period failed to increase the number of v-mPS observed after 24 h post-reactivation, further suggesting they are fully mature and have entered a gel-like state at later stages of replication. When canonical paraspeckles in HeLa cells were treated with 1,6HD, they demonstrated a liquid-like state, with dissolution and recovery within 30 min as previously observed (Fig. S2P)[46,47]. Taken together, these results highlight that v-mPS show distinct properties associated with condensates.

## KSHV ORF11 protein is essential for the formation of v-mPS

Previous results highlight distinct differences in the structure and composition of v-mPS compared to canonical paraspeckles. To analyse these differences in more detail, the proteome composition of v-mPS was compared to canonical paraspeckles and the presence of any KSHV-encoded proteins in these virus-induced structures was interrogated. As previously highlighted, immunofluorescence studies show SFPQ is completely redistributed from a diffuse nuclear staining into v-mPS at 24 h post-KSHV-lytic reactivation (Fig. 1A), therefore affinity pulldowns were performed using anti-SFPQ or control anti-IgG antibodies in latent or 24 h reactivated TREx-BCBL1-RTA cells, coupled with TMT-labelled quantitative mass spectrometry. Negative control (anti-IgG pulldown) protein abundance was subtracted from total abundance for each protein followed by an abundance change and fold change comparison between KSHV latent and lytic samples. As

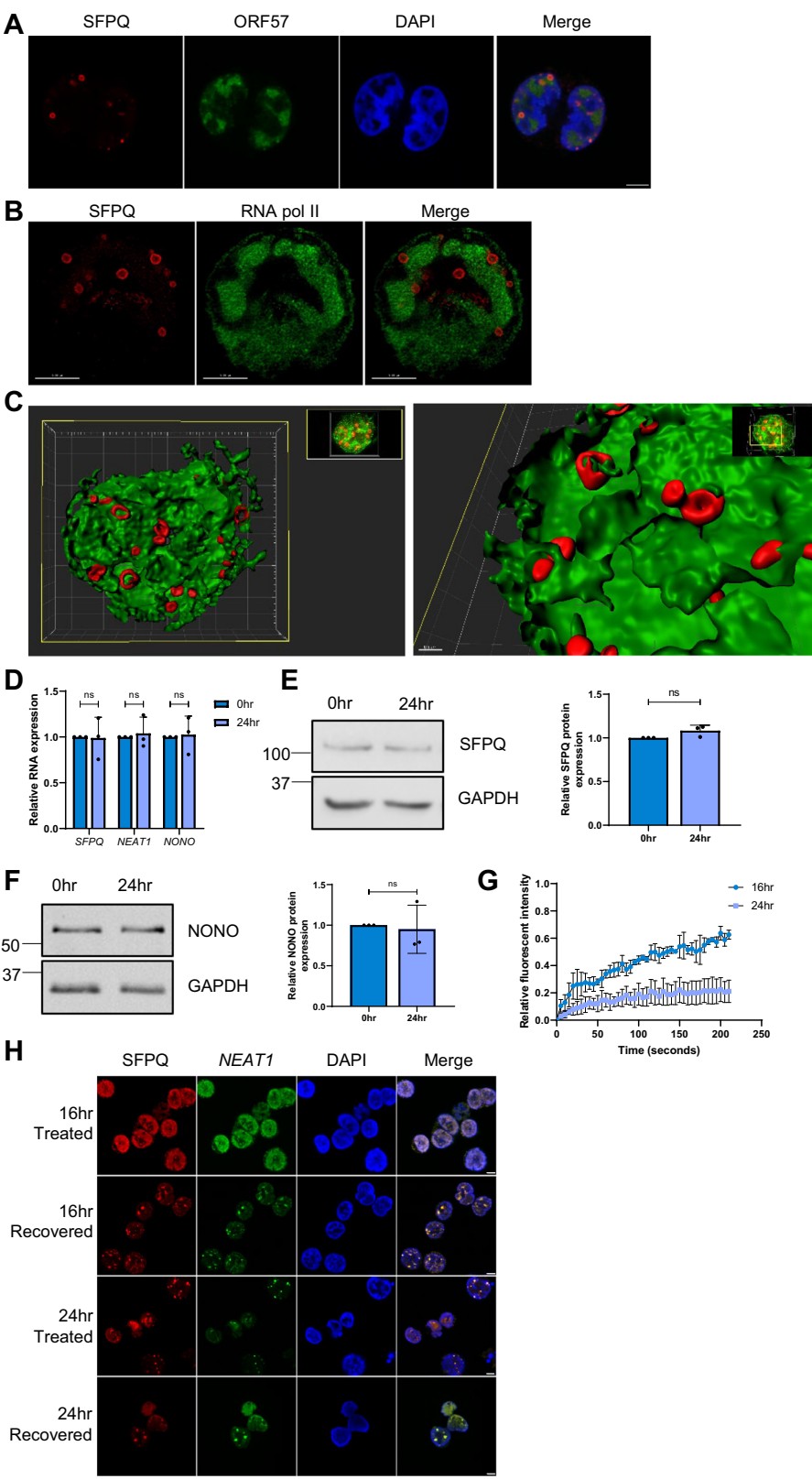

expected, canonical paraspeckle proteins were observed in the SFPQ interaction profile during lytic replication (Fig. S3A). However, interestingly, a selection of DEAD/DEAH box helicases, hnRNPs and the viral ORF11 protein were also identified as interactors with SFPQ exclusively during lytic replication (Fig. 3A). These results suggest these proteins are recruited into v-mPS and may play a role in v-mPS function during KSHV lytic replication.

Identification of the KSHV ORF11 protein prompted the hypothesis that ORF11 may be the virally encoded factor driving the formation of these v-mPS. Like many viral proteins, ORF11 appears to be multifunctional in nature. Previous research has demonstrated that ORF11 is expressed early post-reactivation and has been implicated in the formation of specialised ribosomes[48]. Therefore, to investigate whether KSHV ORF11 was associated with v-mPS, FLAG and SFPQ

**Fig. 2 | Puncta are characterised as virus-induced paraspeckle-like condensates. A** IF of SFPQ (red), ORF57 (green) and DAPI (blue) in 24 h reactivated TREx-BCBL1-RTA cells. **B** STED microscopy for SFPQ (red) and RNA pol II (green) in 24 h reactivated TREx-BCBL1-RTA cells. Deconvolution was performed using Huygens software. **C** Three-dimensional image acquisition was performed using STED microscopy for SFPQ (red) and RNA pol II (green) 24 h post-lytic reactivation KSHV in TREx-BCBL1-RTA cells. Deconvolution was performed in Huygens software before surface rendering of SFPQ and RNA pol II signal using Imaris software. Still images were taken following Z-slicing to reveal the arrangement of SFPQ and RNA pol II. **D** qPCR analysis of *SFPQ*, *NEAT1* and *NONO* levels in TREx-BCBL1-RTA cells with latent KSHV or 24 h post-lytic-KSHV induction, with GAPDH used as a housekeeper (*n* = 3). *P* values are 0.9387, 0.7233 and 0.8378. **E, F** Western blot and densitometry analysis of SFPQ and NONO levels in TREx-BCBL1-RTA cells with latent KSHV or 24 h post-lytic-KSHV induction. GAPDH was used as a loading control (*n* = 3). *P* values are 0.0832 and 0.7845. **G** FRAP analysis of puncta at 16 and 24 h post-KSHV-lytic induction, measuring fluorescent recovery in TREx-BCBL1-RTA SFPQ-GFP OE cells (*n* = 3). **H** IF of TREx-BCBL1-RTA cells at 16 and 24 h post-lytic induction, stained for SFPQ (red), *NEAT1* (green) and DAPI (blue). Cells were treated with 1,6HD and either fixed immediately or allowed to recover for 30 min. All scale bars are 5 μm except C where left image represents 3 μm and right image represents 0.5 μm. In (**D**−**G**) data are presented as mean ± SD. Unpaired two-tailed Student's *t* test was used. Source data are provided as a source data file. All '*n*' repeats are biological in Fig. 2. **P* < 0.05, ***P* < 0.01 and ****P* < 0.001.

co-immunoprecipitations were initially performed in a TREx-BCBL1-RTA FLAG-ORF11 overexpression cell line, due to the lack of an available ORF11 antibody, which confirmed an association between ORF11 and SFPQ proteins. However, this interaction was reduced upon RNase treatment, suggesting that RNA interactions may play a partial role in enhancing the ORF11-SFPQ association (Fig. 3B, C, Fig. S3B). Moreover, pulldowns using in vitro transcribed/translated proteins demonstrated a potential weak association between ORF11 and SFPQ, however, this was enhanced in the presence of NONO, which suggests that ORF11 may directly interact with the SFPQ/NONO heterodimer (Fig. S3C). Together this suggests that ORF11 is capable of binding SFPQ/NONO but the interaction may be enhanced in vivo by the presence of RNA and other potential factors. Immunofluorescence studies coupled with z-stack analysis in the TREx-BCBL1-RTA FLAG-ORF11 cell line also confirmed that ORF11 and SFPQ co-localise in v-mPS at 24 h post-reactivation throughout the nucleus in all z planes (Fig. 3D, Fig. S3D). To further assess ORF11 co-localisation during v-mPS formation, immunostaining was performed during the early stages of lytic replication at 8 and 16 h. Results confirmed that at 16 h post-lytic induction, ORF11 is observed co-localising with immature v-mPS, as shown through co-localisation with SFPQ (Fig. 3E).

To determine whether KSHV ORF11 was required for v-mPS formation, a previously characterised CRISPR/Cas9 TREx-BCBL1-RTA ORF11 knockout cell line was utilised[48]. As expected, immunofluorescence studies confirmed that SFPQ was not redistributed into puncta or any v-mPS formed in ORF11 knockout cells upon reactivation compared to a scrambled cell line, however early vRTCs were still evident, further supporting a direct role for ORF11 in the biogenesis of v-mPS (Fig. 4A).

A classical trait of many paraspeckle proteins is the presence of intrinsically disordered regions (IDRs), which help in the formation of paraspeckles by driving phase separation. Notably, IDRs are subject of extensive reversible post-translational modifications (PTMs), such as phosphorylation, methylation and glycosylation, which regulate changes in their structural and dynamic properties[49,50]. Among these, phosphorylation is one of the most common and important PTMs[51–53]. Bioinformatic studies have shown that phosphorylation of IDRs or intrinsically disordered proteins (IDPs) occurs more frequently than that of folded proteins[54]. This could be due to their increased flexibility and therefore improved kinase accessibility. Structural analysis prediction software (PrDOS and FuzDrop) identified 2 putative IDRs at the N- and C- termini of ORF11 (Fig. S3E). Moreover, PTM prediction software (NetPhos3.1, MusiteDeep) identified 5 potential serine phosphorylation sites within these IDRs, in contrast, no other potential post-translational modifications were predicted (Fig. S3E). Consequently, both termini were deleted in a dual ORF11 truncation mutant (ΔIDR ORF11), and the 5 serine sites were replaced by alanines in a serine/alanine ORF11 mutant (ΔSerine5 ORF11) (Fig. S3E). These mutant ORF11 constructs were used alongside wildtype ORF11 in rescue studies performed in ORF11-CRISPR TREx-BCBL1-RTA cells to discern whether the ORF11 with modified IDRs could still maintain the ability to drive v-mPS formation, identified by both SFPQ and *NEAT1* staining.

Overexpression of ΔIDR ORF11 failed to rescue v-mPS formation, in contrast to overexpression of wildtype ORF11 construct (Fig. 4A, Fig. S3F). Interestingly, expression of the serine mutant, ΔSerine5 ORF11, led to a partial rescue of condensate formation, with smaller puncta observed (Fig. 4A, Fig. S3F). The effect of the ORF11 mutants on KSHV lytic replication was further assessed via analysing the levels of the viral proteins ORF57 and ORF65, with only the full-length expression ORF11 capable of rescue for ORF65 levels (Fig. 4B). Taken together, these rescue studies implicate a role for the ORF11 IDRs in driving v-mPS which is required for KSHV lytic replication. Moreover, it suggests that serine residues may play at least a partial role in condensate formation likely due to their putative phosphorylation capabilities. The predicted propensity of ORF11 to drive v-mPS formation due to phase separation was further assessed by determining whether recombinant ORF11 protein (Fig. S3G) was sufficient to drive droplet formation in vitro. After incubation with 10% PEG for 2 h, recombinant ORF11 was observed to undergo phase separation allowing the formation of droplets, whereas, the control GST tag protein failed to phase separate (Fig. 4C). This was confirmed through Flow Induced Dispersion Analysis (FIDA), where recombinant protein was injected into a capillary, if the protein is capable of LLPS and forms droplets, the detector will record a signal spike, enabling accurate quantification of a protein's ability to form droplets. Here FIDA analysis confirmed the ability of ORF11 to form droplets through recording multiple such signal spikes (Fig. S3H)[55,56]. Taken together, these results suggest a potential function of ORF11 driving v-mPS formation through its capability to phase separate and form droplets.

Specifically, these results suggest ORF11 may have a scaffolding role in v-mPS formation, similar to that of FUS in canonical paraspeckles. Notably, the absence of FUS from v-mPS leads to the hypothesis that ORF11 may act as a FUS replacement. To investigate this further, ORF11 was expressed in FUS-depleted HeLa cells which have previously been shown to be incapable of condensate formation (Fig. S2J). RNA FISH and immunofluorescence was then performed, staining for SFPQ and *NEAT1* to determine whether ORF11 could rescue paraspeckle formation in the absence of FUS. Upon ORF11 overexpression in FUS-depleted HeLa cells, SFPQ and *NEAT1* were shown to co-localise in puncta, suggesting ORF11 is capable of rescuing condensate formation in the absence of FUS (Fig. S3I). Interestingly, the SFPQ/*NEAT1*/ORF11 puncta in rescued HeLa cells were larger than the canonical paraspeckles seen in wildtype HeLa and similar in diameter to v-mPS observed in TREx-BCBL1-RTA cells, up to 2 μm in diameter compared to 0.5 μm respectively (Fig. S3I).

Due to the specificity of KSHV ORF11 localising to the v-mPS, affinity pulldowns coupled to TMT-labelled quantitative mass spectrometry were repeated using anti-FLAG or control anti-IgG antibodies in 24 h reactivated TREx-BCBL1-RTA FLAG-ORF11 overexpressing cells (due to a lack of ORF11-specific antibodies) (Fig. 4D, Fig. S3J). The interaction profiles were interrogated for the presence of essential and core paraspeckle proteins, not surprisingly all early paraspeckle biogenesis factors were present, namely SFPQ, NONO and hnRNP K, indicating ORF11 could play a role in early v-mPS formation[5]. ORF11

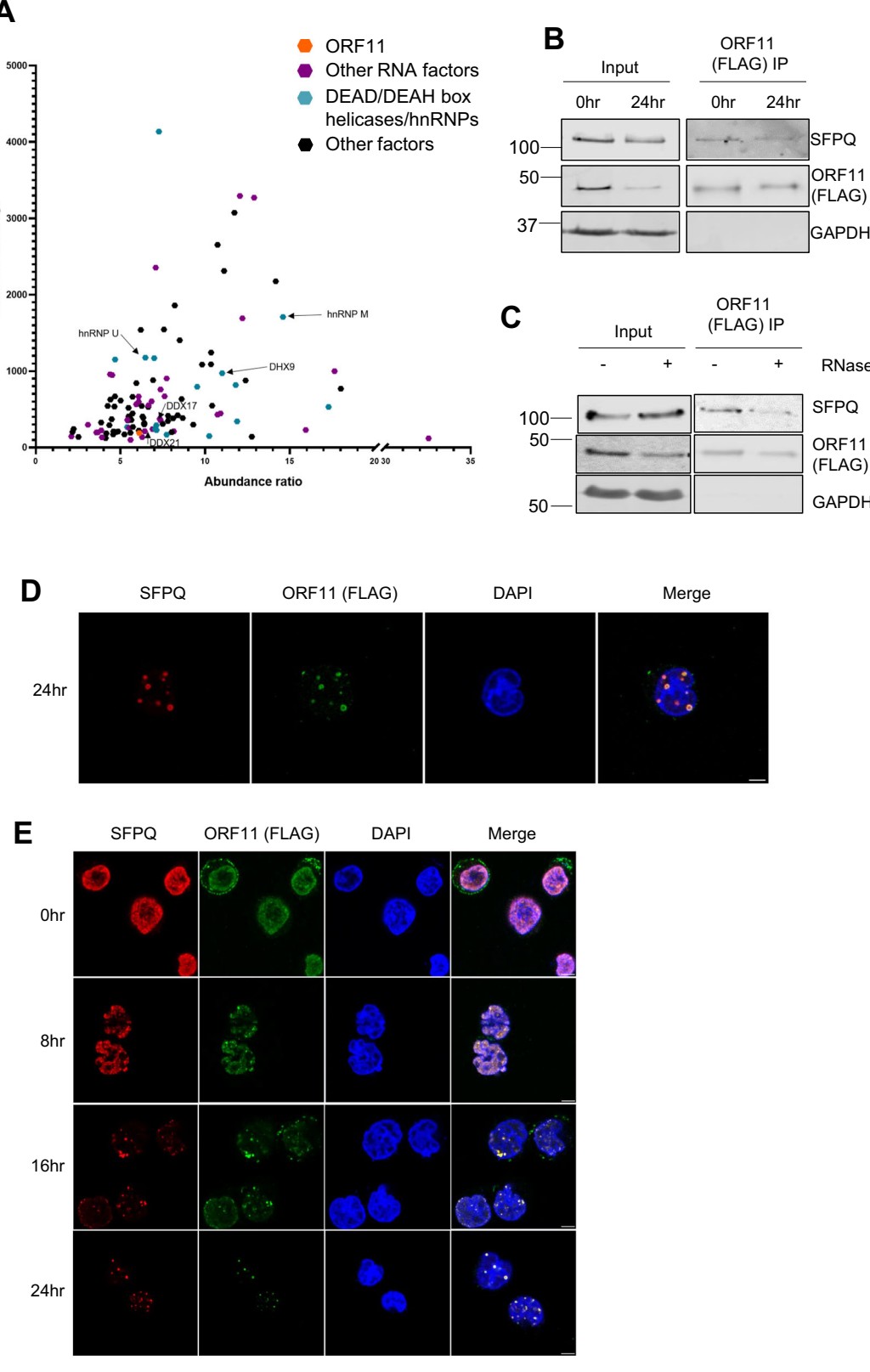

**Fig. 3 | The viral protein ORF11 associates with v-mPS. A** Graph of proteins over a 10% abundance ratio and 4% abundance change in the SFPQ TMT-MS at 24 h (*n* = 2). ORF11 is highlighted in orange, proteins involved in RNA processing are highlighted purple and DEAD/DEAH box helicases and hnRNP proteins are blue. Proteins confirmed via IF as co-localising are marked with arrows. **B** Western blot analysis of FLAG Co-IPs in TREx-BCBL1-RTA FLAG-ORF11 O/E cells during latent or 24 h post-lytic replication and probed with antibodies against FLAG (ORF11), SFPQ, ORF57 and GAPDH (*n* = 3). **C** Western blot analysis of FLAG Co-IPs in TREx-BCBL1-RTA FLAG-ORF11 O/E cells at 24 h ± RNase. Probed with antibodies against FLAG (ORF11), SFPQ and GAPDH. **D** IF for SFPQ (red), FLAG (ORF11) (green) and DAPI (blue) after 24 h of lytic KSHV replication in TREx-BCBL1-RTA FLAG-ORF11 O/E cells. **E** IF of FLAG (ORF11) (green), SFPQ (red) and DAPI (blue) at 0, 8, 16 and 24 h in TREx-BCBL1-RTA FLAG-ORF11 O/E cells harbouring latent KSHV or 8, 16, and 24 h post-lytic induction. Scale bars are 5 μm. All '*n*' repeats are biological in Fig. 3. Source data are provided as a source data file.

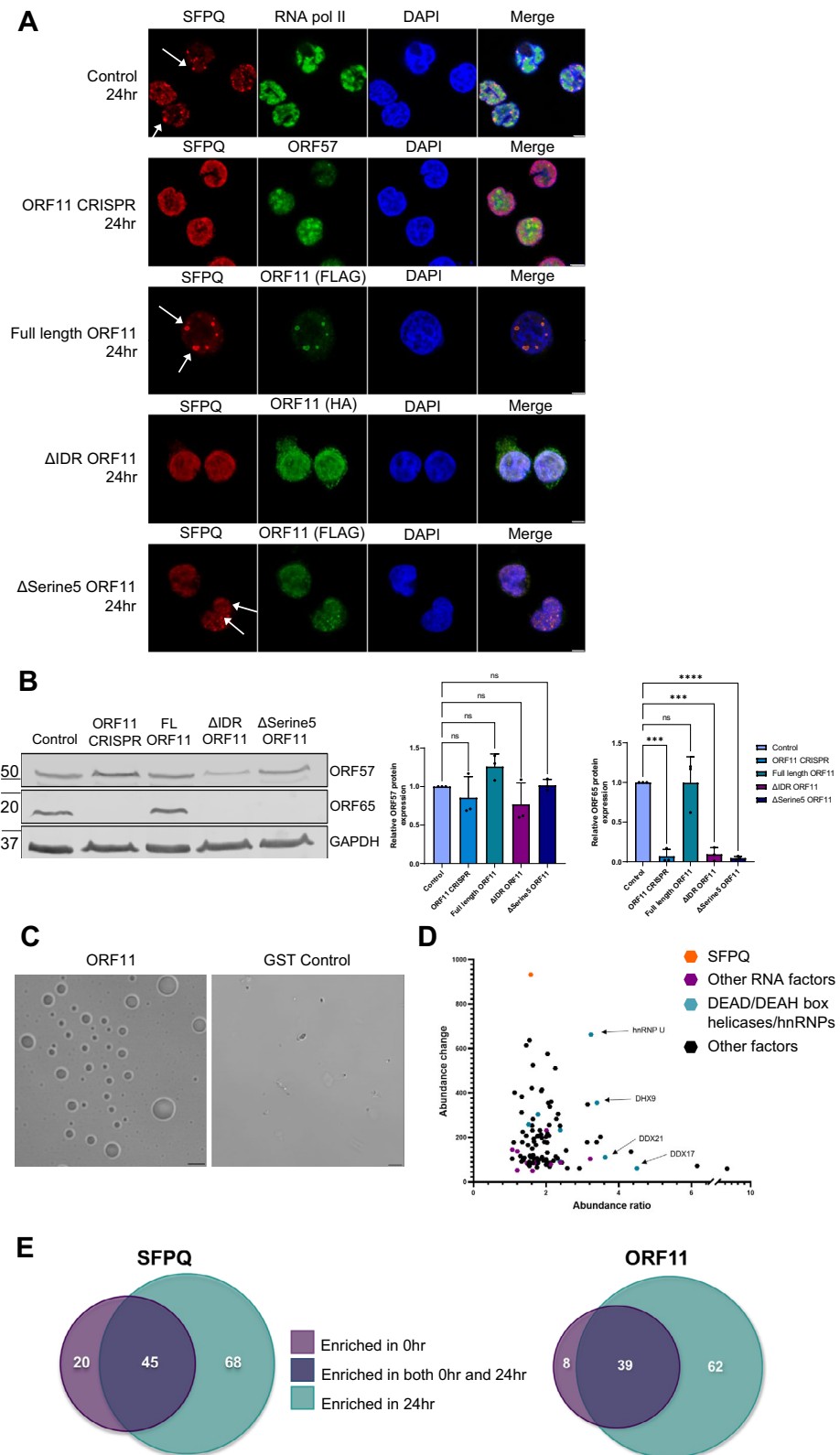

affinity pulldowns also reinforced the interactions observed in the SFPQ affinity pulldowns, highlighting an enrichment of multiple RNA processing factors not commonly reported to associate with canonical paraspeckles, in particular members of the hnRNP and DEAD/DEAH box helicase families, such as DDX17, DDX21, DDX3X, DDX5, DHX9, hnRNP U (Fig. 4D, Fig. S3K). Venn diagrams (Fig. 4E) highlight the shift in interaction profiles of both SFPQ and ORF11 between latent and lytic

replication, suggesting a distinct role during lytic replication, with 68 and 62 new enriched protein interactors for SFPQ and ORF11, respectively at 24 h compared to 0 h. The number of new proteins at 24 h is higher for both bait proteins than the number of conserved interactors found at both time points (45 and 39, respectively), again highlighting the shift in protein interaction during viral replication. A selection of these proteins highlighted in the TMT-MS (DHX9, hnRNP M, DDX17,

**Fig. 4 | ORF11 is the viral driver in condensate formation. A** IF staining of control TREx-BCBL1-RTA cells (Row 1), ORF11 CRISPR TREx-BCBL1-RTA cells (row 2), full length ORF11 rescue TREx-BCBL1-RTA (row 3), truncated ORF11 rescue TREx-BCBL1-RTA (ΔIDR ORF11) (row 4) or serine mutated ORF11 rescue TREx-BCBL1-RTA (ΔSerine5 ORF11) (row 5). Staining is for SFPQ (red), DAPI (blue) and either RNA pol II, ORF57, FLAG or HA (green). White arrows are used to highlight v-mPS. All cells were at 24 h post-reactivation. **B** Representative western blot of ORF57 and ORF65 in control, ORF11 CRISPR, full-length ORF11 rescue, ΔIDR ORF11 or ΔSerine5 ORF11 TREx-BCBL1-RTA cells at 48 h post-KSHV lytic reactivation. GAPDH was used as a loading control. Densitometry analysis of ORF57 and ORF65 was done on $n = 3$. ORF57 densitometry $P$ values are 0.7683, 0.3346, 0.4309 and 0.9999. $P$ values for ORF65 densitometry are 0.0001, >0.9999, 0.0001 and <0.0001. **C** Droplet propensity assay. Purified ORF11 or purified GST alone was combined with 10% PEG and incubated for 2 h before visualisation by bright field microscopy with DIC. **D** Graph of proteins over a 10% abundance ratio and 4% abundance enriched in the FLAG-ORF11 TMT-MS at 24 h ($n = 2$). SFPQ is highlighted in orange, proteins involved in RNA processing are highlighted purple and DEAD/DEAH box helicases and hnRNP proteins are blue. Proteins confirmed via IF as co-localising are marked with arrows. **E** Venn diagrams showing the change and overlap in the protein interactome of SFPQ (left) and ORF11 (FLAG) (right) between the latent conditions (purple) and 24 h lytic replication conditions (green) from co-immunoprecipitations coupled with TMT-LC-MS. Proteins interacting only during latency are counted in purple section, only interacting in lytic replication are counted in green section and proteins interacting in both are represented in the blue overlapping section. In (**B**) data are presented as mean ± SD. One-way ANOVA was used. All '$n$' repeats are biological in Fig. 4. *$P < 0.05$, **$P < 0.01$ and ***$P < 0.001$. Source data are provided as a source data file.

DDX21 and hnRNP U) were confirmed to co-localise with v-mPS using immunofluorescence studies (Fig. S3L–P), however, it must be noted that in some cases only a proportion of the protein re-localised to v-mPS, with some of the factors also having a generalised nucleoplasm localisation. Together this data suggests that the KSHV ORF11 protein co-localises to and is required for the formation of v-mPS. Interestingly, we observed a lack of paraspeckle formation in latent ORF11 overexpressing TREx-BCBL1-RTA cells, with localisation of ORF11 to the cellular membrane, suggesting that additional viral lytic factors may be required to localise ORF11 to the nucleus or help to drive v-mPS biogenesis. Additionally, there are distinct proteomic interactions highlighted in both SFPQ and ORF11 TMT-MS that have not yet been reported in canonical paraspeckles, with some of these factors potentially aiding in biogenesis or novel functions of v-mPS.

## v-mPS formation is required for KSHV lytic replication

SFPQ is an essential component of canonical paraspeckles and is exclusively redistributed into v-mPS at 24 h post-reactivation, therefore loss of SFPQ is predicted to abrogate v-mPS formation and is an ideal target to assess the importance of v-mPS formation for virus replication[43]. To test this hypothesis, stable SFPQ knockdown cell lines were produced in TREx-BCBL1-RTA cells transduced with targeted shRNAs. Effective SFPQ knockdown reduced SFPQ mRNA levels by 60%, and protein levels by 70%, respectively (Fig. 5A, Fig. S4A, B). The effect of SFPQ depletion on v-mPS formation was then assessed compared to a scrambled control using NONO-specific antibodies, as the paraspeckle marker. Results confirmed that v-mPS failed to form in SFPQ-depleted cells compared to control cells (Fig. 5B). These cell lines were then utilised to assess what impact the loss of SFPQ and the resulting failure to produce v-mPS had upon KSHV lytic replication. Reactivation of these v-mPS deficient SFPQ-depleted cells showed a dramatic reduction, of at least 90%, in both the early ORF57 and the late ORF65 mRNA and protein levels (Fig. 5C–E). To confirm that SFPQ depletion affected KSHV lytic replication, viral load and infectious virion production were also assessed compared to scramble control cells. Viral genomic DNA was measured via qPCR from scrambled and SFPQ-depleted TREx-BCBL1-RTA cells and results showed SFPQ depletion led to a 70% reduction compared to scrambled control (Fig. 5F). In addition, supernatants of reactivated scrambled and SFPQ-depleted TREx-BCBL1-RTA cells were used to re-infect naïve HEK-293T cells and infectious virion production quantified by qPCR. Cells re-infected with supernatant from SFPQ-depleted cells resulted in a dramatic loss of infectious virions, reduced by 80% compared to controls (Fig. 5G). Taken together, these results suggest that SFPQ depletion and the resulting failure to form v-mPS significantly impacts KSHV lytic replication and infectious virion production.

However, SFPQ is a multi-functional protein, having various roles in regulating gene expression in the nucleus, therefore to further determine whether KSHV lytic replication only requires SFPQ or requires other components associated with v-mPS, we first assessed the effect of depleting NONO (Fig. 5H, Fig. S4C, D), which again showed a significant reduction in viral replication as analysed through ORF65 protein expression, with a 50–70% decrease, however in contrast to the SFPQ depleted cell line ORF57 levels were not affected (Fig. 5I, J). IF analysis of NONO KD cells showed the majority of lytic cells failed to form v-mPS, and any that did form were significantly smaller and misshapen compared to the scrambled control (Fig. S4E). Furthermore, when *NEAT1* expression was depleted by GapmeRs (antisense complementary oligonucleotides that induce target degradation through RNAse H recruitment), v-mPS once again failed to form in lytic TREx-BCBL1-RTA cells, which resulted in a significant drop in ORF65 protein expression of approximately 75%, but again no reduction in ORF57 levels was observed (Fig. S4F–H). Interestingly cells depleted of ORF11, NONO or *NEAT1* had similar deleterious effects on KSHV, with reductions in the level of late viral proteins, whereas loss of SFPQ led to a more dramatic phenotype, with a reduction in early viral gene expression. It is likely that this is due to a multifunctional role of SFPQ, whereas the effects observed with loss of the other factors, ORF11, NONO or *NEAT1*, are likely due to loss of v-mPS. Together these findings suggest that v-mPS formation is required for KSHV lytic replication.

## v-mPS and their associated PSPs are involved in RNA processing

The dynamic nature of paraspeckle protein and RNA composition, aligned with the fact that PSPs are generally multifunctional in nature and not exclusively confined to paraspeckles, has complicated the elucidation of paraspeckle function. However, due to the enrichment of RNA helicases and hnRNP members within the v-mPS and their adjacent localisation to vRTCs, we hypothesised that the v-mPS may play a role in viral RNA processing events during KSHV lytic replication. To examine whether viral transcripts are associated with the v-mPS, we utilised the biotin-based proximity labelling technique HyPro-Seq[57,58] in reactivated TREx-BCBL1-RTA cells. *NEAT1* was used as the probe target, as it predominantly localises to the v-mPS during KSHV lytic replication, and all RNAs within a 20 nm distance of *NEAT1* were biotinylated, isolated and identified via next-generation sequencing (NGS). Analysis identified an enrichment of KSHV transcripts in the *NEAT1* interactome and when mapped, many cluster within regions of the viral genome (Fig. 6A). Upon further analysis, 40% of the viral transcripts are spliced, 80% are polycistronic transcripts, and 90% are $m^6A$ methylated correlated against previous KSHV genome and $m^6A$ datasets (Fig. 6B)[24,59]. Taken together, the abundance of viral transcripts alongside the enrichment of RNA processing proteins observed in the TMT-MS, further support a potential role of v-mPS in viral RNA processing. To validate the HyPro-Seq dataset, SFPQ and FLAG-ORF11 RNA immunoprecipitations (RIPs) coupled with qPCR were performed and results showed that the majority of viral transcripts identified by HyPro-Seq were precipitated, confirming the association of viral transcripts with the v-mPS components, SFPQ and ORF11 (Fig. 6C, D). Again, these results suggest that v-mPS may have a key role

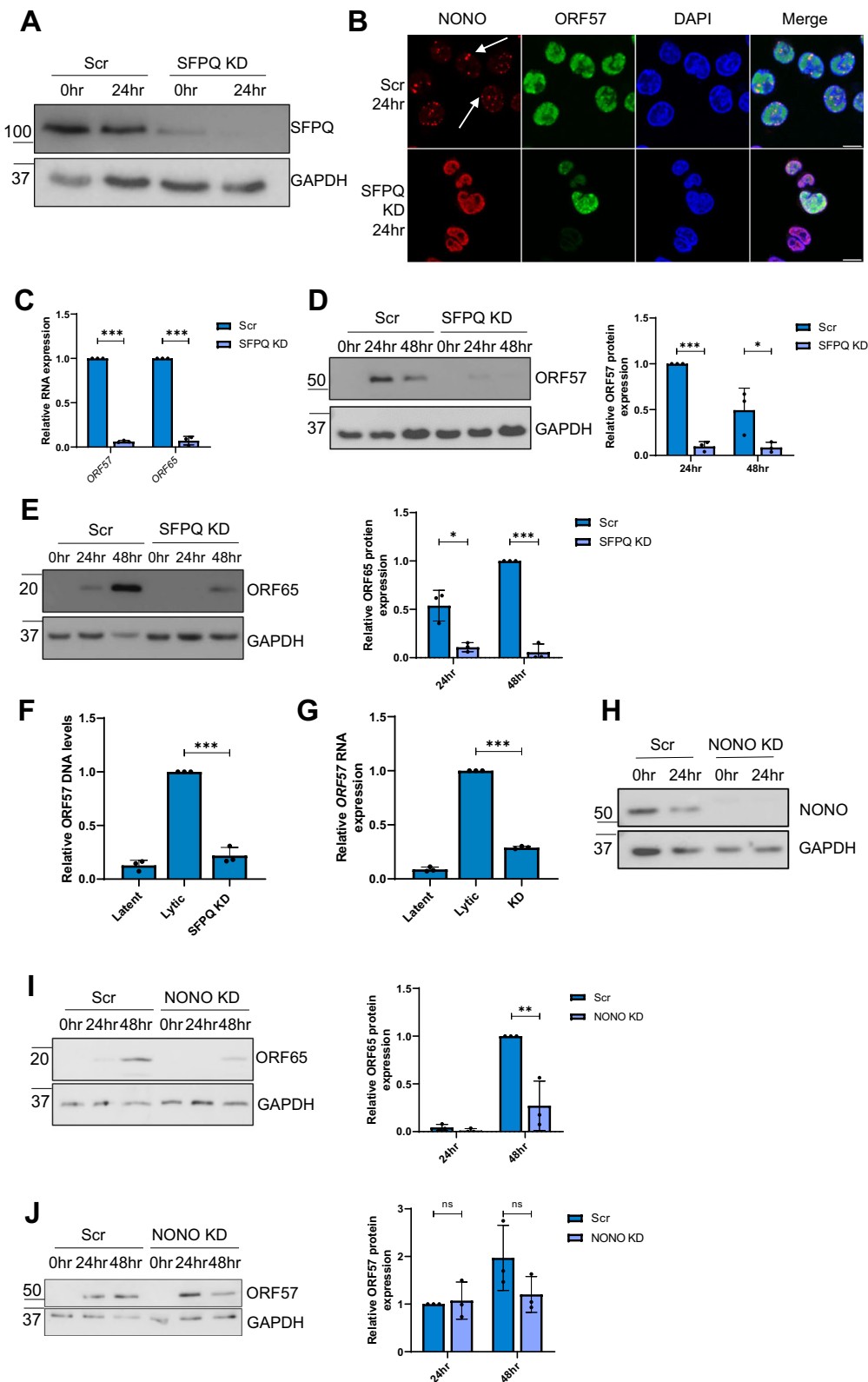

in viral RNA processing, however, v-mPS are localised around vRTCs, which are also implicated in genome replication. Therefore, to determine whether viral genomes were also associated with v-mPS, DNA FISH was used to stain viral replicating DNA using viral DNA-specific probes to ORF50. Results showed that viral genomes were only observed in vRTCs and were absent from v-mPS, suggesting v-mPS are not associated with viral genome replication (Fig. S5A). Moreover, to

assess viral transcription initiation, dual luciferase reporter constructs containing the 5' UTRs of early and late viral ORFs were transfected into SFPQ overexpression HEK-293T cell lines and wildtype HEK-293T cells, and luciferase activity was measured. The SFPQ overexpression cell line showed no increase in promoter activity and transcription of viral ORFs compared to the wildtype HEK-293T cells (Fig. S5B). Taken together, these results suggest v-mPS do not have a role in DNA

**Fig. 5 | v-mPS are essential for viral replication. A** Representative western blot of SFPQ levels in TREx-BCBL1-RTA scr and TREx-BCBL1-RTA SFPQ KD cell lines with GAPDH as a loading control (*n* = 3). **B** IF analysis in TREx-BCBL1-RTA scr and TREx-BCBL1-RTA SFPQ KD cell lines with staining against NONO (red), ORF57 (green) and DAPI (blue). White arrows are used to highlight v-mPS. **C** qPCR analysis of *ORF57* and *ORF65* levels in TREx-BCBL1-RTA scr and TREx-BCBL1-RTA SFPQ KD TREx cells after 24 h of KSHV lytic replication. GAPDH was used as a housekeeper (*n* = 3). *P* values are <0.0001 and <0.0001. **D**, **E**. Representative western blots of ORF57 (**D**) and ORF65 (**E**) levels in TREx-BCBL1-RTA scr and TREx-BCBL1-RTA SFPQ KD cell lines with GAPDH as a loading control. Densitometry analysis of (*n* = 3) for ORF57 and ORF65. *P* values are <0.0001 and 0.0461. **F** qPCR analysis of ORF57 DNA levels for viral load at 72 h post-KSHV induction, with TREx-BCBL1-RTA scr and TREx-BCBL1-RTA SFPQ KD including latent cells as a reactivation control and GAPDH as a housekeeper (*n* = 3). *P* value is <0.0001. **G** qPCR analysis of *ORF57* in HEK-293 T cells for reinfection assay (*n* = 3). *P* value is <0.0001. **H** Representative western blot of NONO protein levels in scr and NONO KD TREx cells with GAPDH as a loading control. **I** Representative western blot of ORF65 protein levels in TREx-BCBL1-RTA scr and TREx-BCBL1-RTA NONO KD cell lines with GAPDH as a loading control. Densitometry analysis of *n* = 3. *P* value is 0.0081. **J** Representative western blot of ORF57 protein levels in TREx-BCBL1-RTA scr and TREx-BCBL1-RTA NONO KD cell lines with GAPDH as a loading control. Densitometry analysis of *n* = 3. *P* values are 0.7613 and 0.1635. Scale bars in B are 10 μm. In C-G, I-J, data are presented as mean ± SD. *\**P* < 0.05, \*\**P* < 0.01 and \*\*\**P* < 0.001 (unpaired two-tailed Student's *t* test). All '*n*' repeats are biological in Fig. 5. Source data are provided as a source data file.

replication or transcription initiation, further supporting the hypothesis that their primary role may be in RNA processing.

To further investigate the potential function of v-mPS on viral mRNA processing, we explored the processing of one of the most enriched viral transcripts in the SFPQ and ORF11 RIPs, namely *K8*. This was selected for further investigation to assess v-mPS functionality due to its well characterised alternative splicing profile[60]. Firstly, RNA-FISH was utilised to confirm that *K8* transcripts were localised to the v-mPS (Fig. 6E). Moreover, SFPQ RIPs were performed in control and ORF11 CRISPR cells to assess the association of *K8* specificity with v-mPS formation. Notably the interaction between SFPQ and *NEAT1* was preserved within the ORF11 CRISPR cells, in contrast, the interaction with *K8* was drastically reduced (Fig. 6F). This reinforces the observation that the interaction between SFPQ and viral transcripts is dependent on v-mPS formation.

The *K8* transcript undergoes alternative splicing to produce three spliced variants, termed α, β and γ. It has been previously demonstrated that the majority of the pre-mRNA γ transcript is processed into the functional α transcript and a small proportion of transcripts retain an intron, becoming the β variant which is not translated[60,61]. Although the exact role of these alternative spliced variants is unknown, maintaining this ratio is critical for KSHV lytic replication, with the β truncated protein potentially acting as an antagonist for translation of the primary K8 protein[60–62]. To elucidate whether v-mPS-associated components, SFPQ and ORF11, have a role in orchestrating *K8* processing, a rudimentary PCR assay was performed to analyse the effect v-mPS formation had on the production of alternative *K8* spliced variants. Here *K8* splicing was assessed in the non v-mPS forming SFPQ-depleted and ORF11 knockout cell lines compared to scrambled controls. Notably, in both cell lines which fail to form v-mPS, enhanced levels of the non-translated *K8* β transcript were observed compared to baseline α levels (Fig. 6G, H, Fig. S5C, D). This implicates loss of v-mPS formation in the dysregulation of viral RNA processing.

## v-mPS associated PSP, SFPQ, enhances circRNA levels during KSHV lytic replication

Recently SFPQ has also been postulated to have a role in circRNA biogenesis[14]. Therefore, aligned with the recent observation that KSHV encodes its own circRNAs (kcircRNAs) and also manipulates host circRNA levels during lytic replication[63,64], a potential association in the biogenesis and localisation of circRNAs to the v-mPS-associated PSP, SFPQ was investigated. Although many kcircRNAs have been identified through sequencing, two of the most abundant are kcircPAN and circvIRF4[63,65,66]. Notably, SFPQ RIPs coupled with qPCR showed a clear association with both kcircRNAs (Fig. 6I) and interestingly their levels were disproportionately downregulated over their parental linear transcripts in SFPQ-depleted cells (Fig. 6J, K). Moreover, to determine whether cellular circRNAs are involved in virus replication, SFPQ RIPs were performed to identify host circRNAs potentially associated with v-mPS and whether their levels were affected by SFPQ depletion. Both

circCDYL and circEYA1 were identified as enriched within SFPQ RIPs coupled with qPCR (Fig. 6L)[12,14]. To determine whether these SFPQ-interacting cellular circRNAs had any specific role in virus replication, qPCR analysis was used to assess any changes in levels comparing latency and lytic time points and results showed that both circCDYL and circEYA1 levels were increased during lytic replication, independently of their parental linear transcripts (Fig. S5E, F). Importantly, shRNA-mediated depletion of either circEYA1 or circCDYL resulted in a significant loss of the late viral ORF65 protein, thus highlighting the importance of these circRNAs to KSHV lytic replication (Fig. S5G). Finally, similar to KSHV-encoded circRNAs, their upregulation was significantly impacted by SFPQ depletion. NONO-depleted cells also confirmed this upregulation was significantly reduced upon loss of NONO (Fig. S5H–K). Together these results suggest that the v-mPS associated PSP, SFPQ, is important for both viral and host cell circRNA biogenesis during lytic replication.

## Large v-mPS are formed in gammaherpesvirus infection
To examine whether the novel enlarged v-mPS identified during KSHV lytic infection are formed during other herpesvirus infections, immunofluorescence studies were performed in human foreskin fibroblast (HFF) cells infected with either the alpha-herpesvirus Herpes simplex virus type 1 (HSV-1) or the beta-herpesvirus human cytomegalovirus (HCMV). RNA FISH using probes against *NEAT1* was performed in HSV-1 and HCMV infected HFF cells, surprisingly no enlarged v-mPS were formed during either infection. Unlike, TREx-BCBL1-RTA cells, in both uninfected and infected cells, RNA FISH revealed the presence of what appeared to be canonical paraspeckles which did not significantly change in size during infection. A small increase in paraspeckle number was observed during HSV-1 infection although these did not increase in size, whereas a more diffuse *NEAT1* staining was observed during HCMV infection (Fig.7A). Additionally, immunostaining with SFPQ-specific antibodies also confirmed a lack of enlarged v-mPS in either HSV-1 or HCMV infected cells (Fig.7B).

To determine whether v-mPS are unique to KSHV, or occur in other gammaherpesvirus infections, immunofluorescence studies were repeated in Epstein Barr Virus (EBV)-infected cells: 293-rJJ-L3s, which harbour a BAC EBV clone (the type 2 EBV Jijoye strain) and were reactivated via co-transfection of the EBV lytic proteins ZTA, RTA and BALF4[67,68]. Similar to KSHV, SFPQ re-localised from a diffuse nuclear staining observed during latency to form puncta that clustered around EBV replication compartments, indicated by Ea-D staining, at 24 and 48 h post-lytic induction (Fig. 7C). This suggests that SFPQ may have several roles in both the latent and lytic EBV lifecycles, as recent findings have shown that SFPQ, but not NONO, helps maintain EBV latency by inducing H1 expression and regulating EBV genomic H1 histone occupancy[69]. To confirm the formation of v-mPS during EBV lytic replication *NEAT1* RNA FISH was also performed. Like KSHV, the SFPQ puncta formed during EBV lytic replication were large non-canonical ring-like structures and RNA FISH confirmed the co-

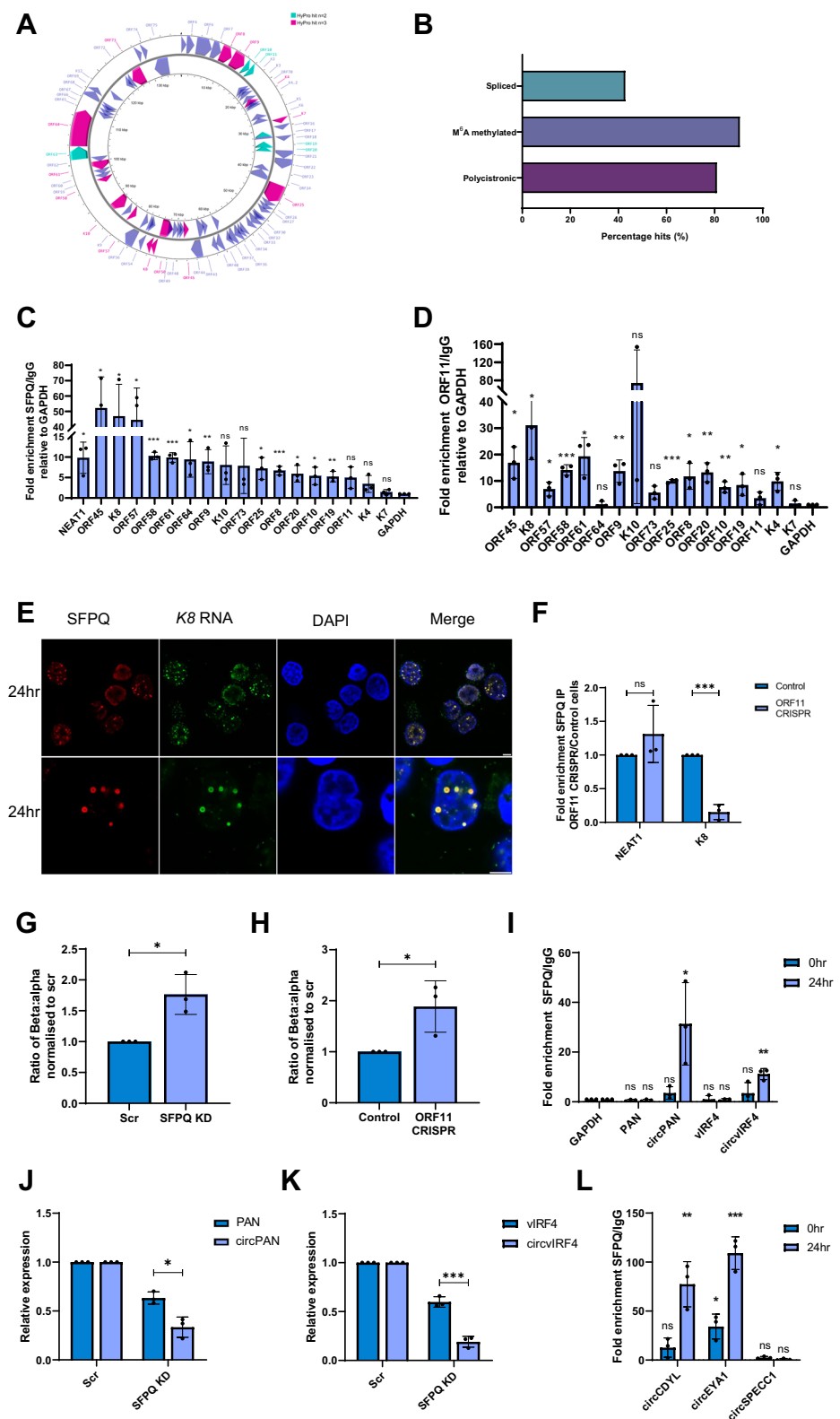

localisation of *NEAT1*, again suggesting the formation of v-mPS (Fig.7D). Immunofluorescence and RNA FISH studies were repeated in a second cell line, 293-SL-HB9-862-L1 which contain the EBV BAC HB9, and results confirmed large puncta, reminiscent of v-mPS surrounding the replication compartments (Fig. S6A, B). These results suggests that v-mPS containing SFPQ and *NEAT1* are only formed during gammaherpesvirus lytic replication cycles.

To assess whether putative EBV v-mPS were formed by an alternative biogenesis pathway to canonical paraspeckles, analogous to KSHV infection, both SFPQ and NONO mRNA and protein expression, as well as *NEAT1* RNA levels, were compared during latent and lytic replication, with levels staying stable during EBV lytic replication, suggesting a non-canonical biogenesis pathway (Fig. 8A, B). To determine whether the v-mPS associated PSP, SFPQ, was also required

**Fig. 6 | v-mPS are implicated in RNA processing. A** KSHV genome with viral ORFs that were enriched in either 2 (green) or 3 (pink) repeats in *NEAT1* Hypro-Seq at 24 h lytic replication. Created in Proksee. **B** Percentage of either *n* = 2 or *n* = 3 viral enriched genes in the Hypro-Seq that are either spliced, m⁶A methylated or from a polycistronic gene. **C** qPCR analysis of SFPQ RIPs for Hypro-Seq hits in TREx at 24 h post-KSHV-lytic induction KSHV showing enrichment over IgG normalised to GAPDH (*n* = 3). *P* values are 0.0161, 0.0119, 0.0184, 0.0222, <0.0001, 0.0002, 0.0282, 0.0099, 0.0613, 0.1539, 0.0157, 0.0008, 0.0138, 0.0237, 0.0046, 0.06, 0.1103 and 0.3317. **D** qPCR analysis of ORF11 (FLAG) RIPs for the Hypro-Seq hits in FLAG-ORF11 OE at 24 h post-KSHV-lytic induction showing enrichment over IgG normalised to GAPDH (*n* = 3). *P* values are 0.0101, 0.0161, 0.0173, 0.0003, 0.0111, 0.8232, 0.0063, 0.1564, 0.0317, <0.0001, 0.0194, 0.004, 0.0053, 0.0333, 0.1560, 0.0111 and 0.5627. **E** IF analysis in 24 h post-lytic-KSHV induction TREx-BCBL1-RTA cells with FISH probes against *K8* (green), antibodies against SFPQ (red) and DAPI staining (blue). Lower row is a more zoomed in field of view. **F** qPCR analysis of SFPQ RIPs in ORF11 CRISPR cells at 24 h post-lytic induction showing enrichment over control cells normalised to NEAT1 (*n* = 3). *P* values are 0.2704 and 0.0002.

**G**, **H** PCR of K8 α, β and γ transcripts in scr and SFPQ KD cells (**G**) or control and ORF11 CRISPR cells (**H**). Ratio of α to β was normalised to scr control. Densitometry analysis performed on *n* = 3. *P* values are 0.0149 and 0.0380. **I** qPCR analysis of SFPQ RIPs in TREx cells at latent and 24 h post-lytic induction showing enrichment over IgG normalised to GAPDH (*n* = 3). *P* values are 0.1231, 0.1877, 0.1579, 0.0338, 0.9584, 0.4785, 0.3683 and 0.0014. **J** qPCR analysis of Pan and circPAN levels in scr and SFPQ KD TREx-BCBL1-RTA cells at 24 h post-lytic induction. GAPDH was used as a housekeeper (*n* = 3). *P* value is 0.0127. **K** qPCR analysis of vIRF4 and circvIRF4 levels in scr and SFPQ KD TREx cells at 24 h post-lytic induction. GAPDH was used as a housekeeper (*n* = 3). *P* value is 0.0008. **L** qPCR analysis of SFPQ RIPs in TREx-BCBL1-RTA cells at latent and 24 h post-lytic induction showing enrichment over IgG normalised to GAPDH (*n* = 3). *P* values are 0.2986, 0.0005, 0.0029, <0.0001, 0.9942 and >0.9999. Scale bars are 5 μm. In (**C**–**G**, **I**–**L**) data are presented as mean ± SD. *\*P* < 0.05, *\*\*P* < 0.01 and *\*\*\*P* < 0.001 (unpaired two-tailed Student's *t* test except for (**L**) (one-way ANOVA). All '*n*' repeats are biological in Fig. 6. Source data are provided as a source data file.

for EBV lytic replication, SFPQ was depleted in the 293-rJJ-L3s cell line and immunofluorescence studies using PSPC1 confirmed that SFPQ was required for v-mPS formation during EBV lytic replication (Fig. 8C). SFPQ depletion also resulted in a 70% decrease in the expression of the EBV lytic protein Ea-D (Fig. 8D) and a reduction in EBV DNA replication, with a 40% reduction in viral load after 72 h of lytic replication (Fig. S6C) suggesting the v-mPS associated PSP, SFPQ, is also required for EBV lytic replication. Finally, 1,6HD treatment experiments showed a maturation process from a liquid-like state during early EBV lytic replication (16 h) to a more gel-like state during the later stages (48 h) (Fig. 8E). Together, these data suggest that large v-mPS are potentially a pan-gammaherpesvirus phenomenon that occur during their lytic replication phase.

## Formation of v-mPS increase genomic instability during infection

The finding that enlarged v-mPS formation was only observed in gammaherpesvirus infection is particularly intriguing and led to further studies exploring a potential link between v-mPS formation and gammaherpesvirus-mediated tumourigenesis. Recent observations have highlighted that both KSHV and EBV lytic replication cycles contribute to an increase in genome instability, although the mechanisms driving virus-mediated tumourigenesis are not fully elucidated[29,70,71]. SFPQ and other PSPs, such as NONO, have recently been implicated in multiple pathways associated with DNA damage, potentially functioning as gatekeepers of genome stability, although whether this is connected to paraspeckle function remains undetermined[41,72–76]. This highlights an intriguing link between the increase in genomic instability and the formation of the v-mPS, which leads to the increased sequestration and association of PSPs, resulting in their depletion from the nucleoplasm. This is particularly evident with SFPQ which shows loss of the diffuse staining throughout the nucleoplasm upon the formation of v-mPS (Fig. 1A).

A key indicator of DNA damage is lagging chromosomes during mitosis, where chromosomes fail to separate into daughter cells. To assess a potential role of SFPQ sequestration into v-mPS in KSHV-mediated genome instability, the presence of chromosomal anomalies was compared in scrambled vs SFPQ depleted latent TREX-BCBL1-RTA or HEK-293T cells. The first 20 mitotic cells identified for each cell line were assessed to minimise bias. Results showed that the percentage of cells with lagging chromosomes increased from 0% to 30%, and from 0% to 50%, respectively upon depletion of SFPQ (Fig. 9A). This implicates SPFQ as a factor in the DDR, supporting the idea that sequestration of SFPQ into v-mPS, which leads to depletion from the nucleoplasm could increase DNA damage.

As previously described, KSHV lytic replication leads to an increase in markers of DNA damage[28], as measured by immunoblotting

using γH2AX-specific antibodies, a marker of double stranded DNA breaks (Fig.9B). To determine whether this increase in DNA damage is linked to SFPQ sequestration into v-mPS, v-mPS formation was disrupted during KSHV lytic replication using propylene glycol (PG). This has a similar mechanism to 1,6HD, allowing disruption of v-mPS in a liquid-like state, but can be used for longer time periods after hyperosmotic adaptation[45]. TREx-BCBL1-RTA cells were therefore incubated with PG between 16–24 h post-lytic induction, leading to an environment in which viral replication could partly occur, however v-mPS formation was abrogated (Fig. S7A). Under these conditions, SFPQ remained in the nucleoplasm and not sequestered to v-mPS. Similarly to NONO, *NEAT1* and ORF11 depletion, treatment with PG led to a reduction in the late KSHV protein ORF65, but not ORF57 (Fig. S7B). To determine whether this reduced SFPQ sequestration due to the lack of v-mPS formation affected levels of genomic instability, DNA damage was compared in KSHV latent and lytically replicating cells, in the absence and presence of PG. Interestingly, in the presence of PG, a significant reduction in γH2AX levels were observed compared to control reactivated cells (Fig. 9C). This suggests that reduced SFPQ sequestration due to v-mPS inhibition may reduce virus-induced genomic instability. To confirm this observation, DNA damage was also compared in the KSHV ORF11 knockout CRISPR TREx-BCBL1-RTA cells, where v-mPS are disrupted but early lytic viral gene expression is still evident. Results show a similar phenotype to PG treatment with reduced γH2AX levels in ORF11 knockdown cells compared to control (Fig. S7C).

To confirm that the increase in DNA damage observed due to v-mPS formation was specifically due to SFPQ sequestration, changes in γH2AX abundance were compared in latent and reactivated TREx-BCBL1-RTA cells either depleted for SFPQ or upon SFPQ overexpression. As expected, depletion of SFPQ resulted in an increase in γH2AX, whereas overexpression of SFPQ, which provides sufficient SFPQ to function in the nucleoplasm as well as being sequestered into v-mPS, led to reduced γH2AX levels regardless of whether KSHV-infected cells remained latent or underwent lytic replication (Fig. 9D, E). Similarly, this increase in DNA damage was also observed during EBV lytic replication, with levels of γH2AX again increasing during lytic replication and inversely correlating with SFPQ levels (Fig. 10A, B, Fig. S7D).

Neutral comet assays were then used to quantify double-stranded breaks in scrambled latent control cells compared to TREx-BCBL1-RTA cells either depleted for SFPQ or upon SFPQ overexpression. A similar phenotype was observed, where SFPQ overexpressing cells displayed the lowest mean olive tail movement (OTM) of 2.8 equating to the least DNA damage, followed by scrambled control cells (8.1), whilst SFPQ KD cells had the greatest mean OTM of 13.1 and therefore the greatest number of dsDNA breaks (Fig. 10C). Together, these results validate

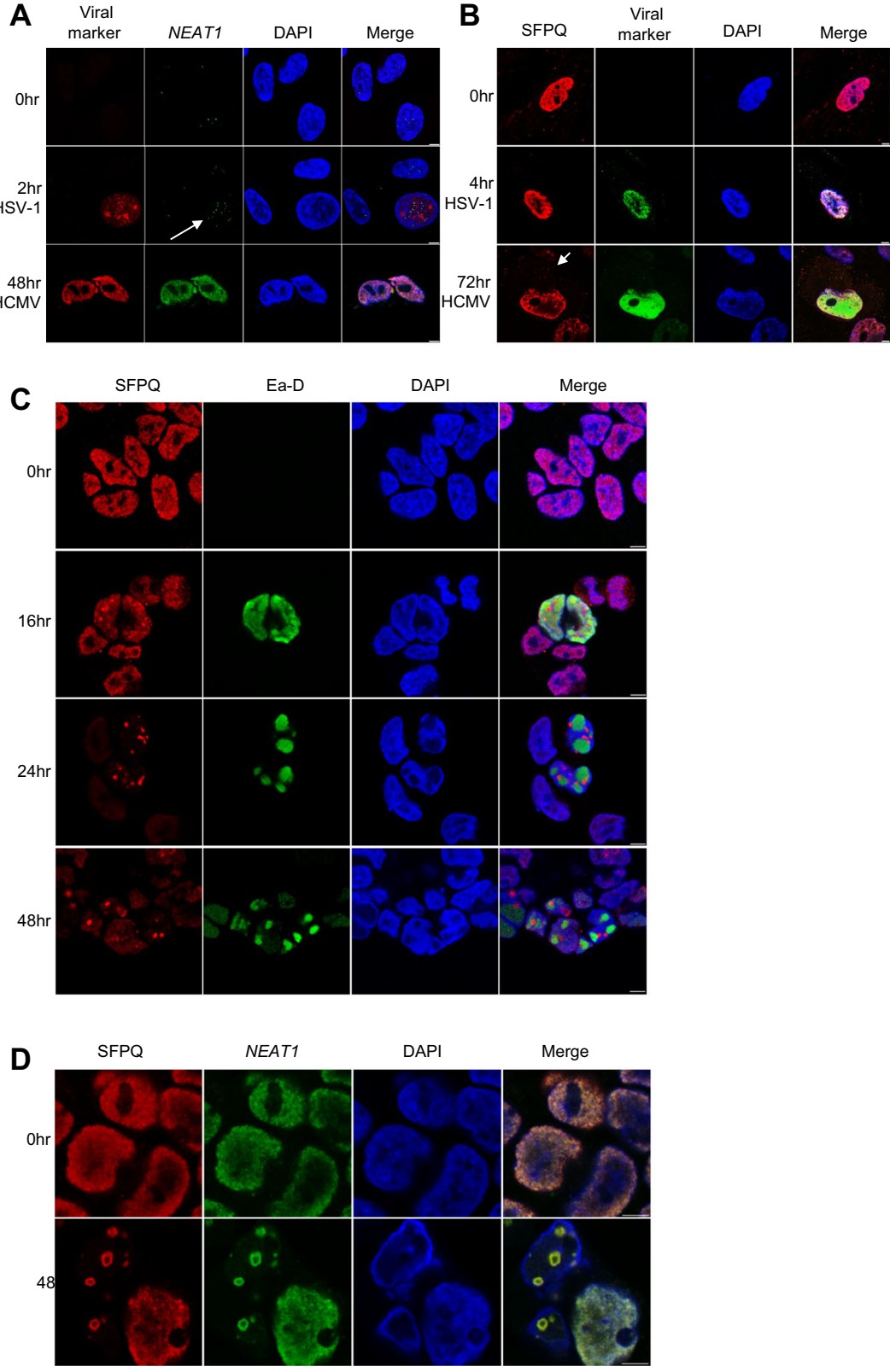

**Fig. 7 | Large v-mPS are only formed in gammaherpesvirus lytic replication. A** IF of HFF cells infected with HSV-1 or HCMV at 0, 2 or 48 h, stained for *NEAT1* (green), DAPI (blue) and either ICP27 for HSV-1 or pp72 for HCMV (red). White arrows are used to highlight increase in *NEAT1* puncta. **B** IF of HFF cells infected with HSV-1 or HCMV at 0, 4 or 72 h, stained for SFPQ (red), DAPI (blue) and either ICP27 for HSV-1 or pp72 for HCMV (green). White arrows are used to highlight more cytoplasmic SFPQ localisation. **C** IF of 293-rJJ-L3s cells at 0, 16, 24 and 48 h post-reactivation with staining against SFPQ (red), Ea-D (green) and DAPI (blue). **D** IF of 293-rJJ-L3s cells at 0 and 48 h reactivation, stained for SFPQ (red), *NEAT1* (green) and DAPI (blue). Scale bars are 5 μm.

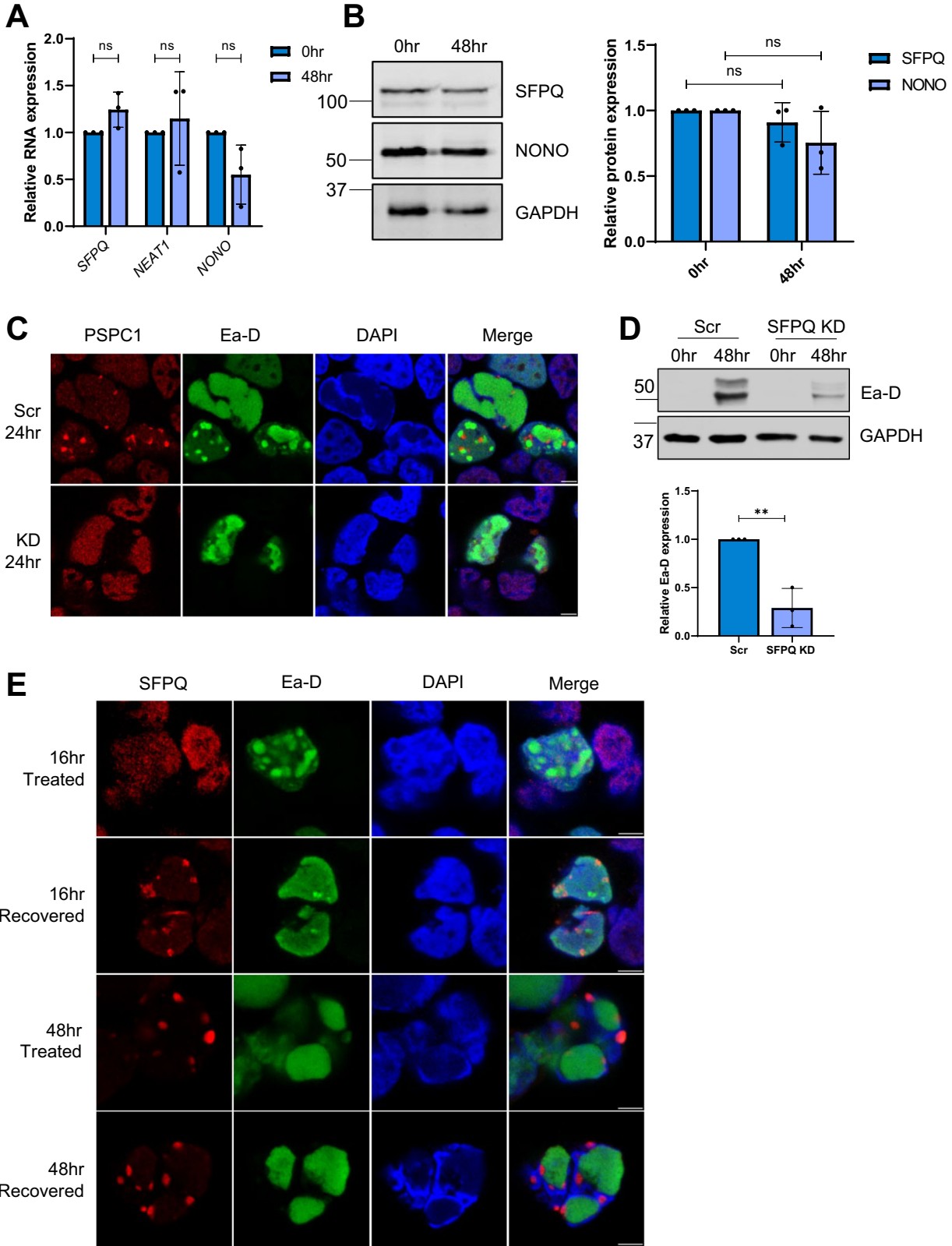

**Fig. 8 | v-mPS formed during EBV lytic replication resemble those formed during KSHV replication. A** qPCR of *SFPQ, NEAT1* and *NONO* levels in rJJ-L3-#1 cells at 0 and 48 h. GAPDH was used as a housekeeper, *n* = 3. *P* values are 0.0876, 0.6313 and 0.0691. **B** Representative western blot of SFPQ and NONO protein levels in 293-rJJ-L3s cells at 0 and 48 h with GAPDH as a loading control. Densitometry analysis performed on *n* = 3. *P* values are 0.3579 and 0.1503. **C** IF analysis in scr and SFPQ KD rJJ-L3-#1 cells with antibodies against PSPC1 (red), Ea-D (green) and DAPI (blue). **D** Representative western blot of Ea-D protein levels in scr and SFPQ KD 293-rJJ-L3s

cells with GAPDH as a loading control. Densitometry analysis of *n* = 3. *P* value is 0.0037. **E** IF of 293-rJJ-L3s cells at 16 and 48 h. Cells were either treated with 1,6 HD or treated and then allowed to recover for 30 min prior to fixing. Cells were stained for SFPQ (red), Ea-D (green) and DAPI (blue). Scale bars are 5 μm. In (**A**, **B**, **D**) data are presented as mean ± SD. \**P* < 0.05, \*\**P* < 0.01 and \*\*\**P* < 0.001 (unpaired two-tailed Student's *t* test). All '*n*' repeats are biological in Fig. 8. Source data are provided as a source data file.

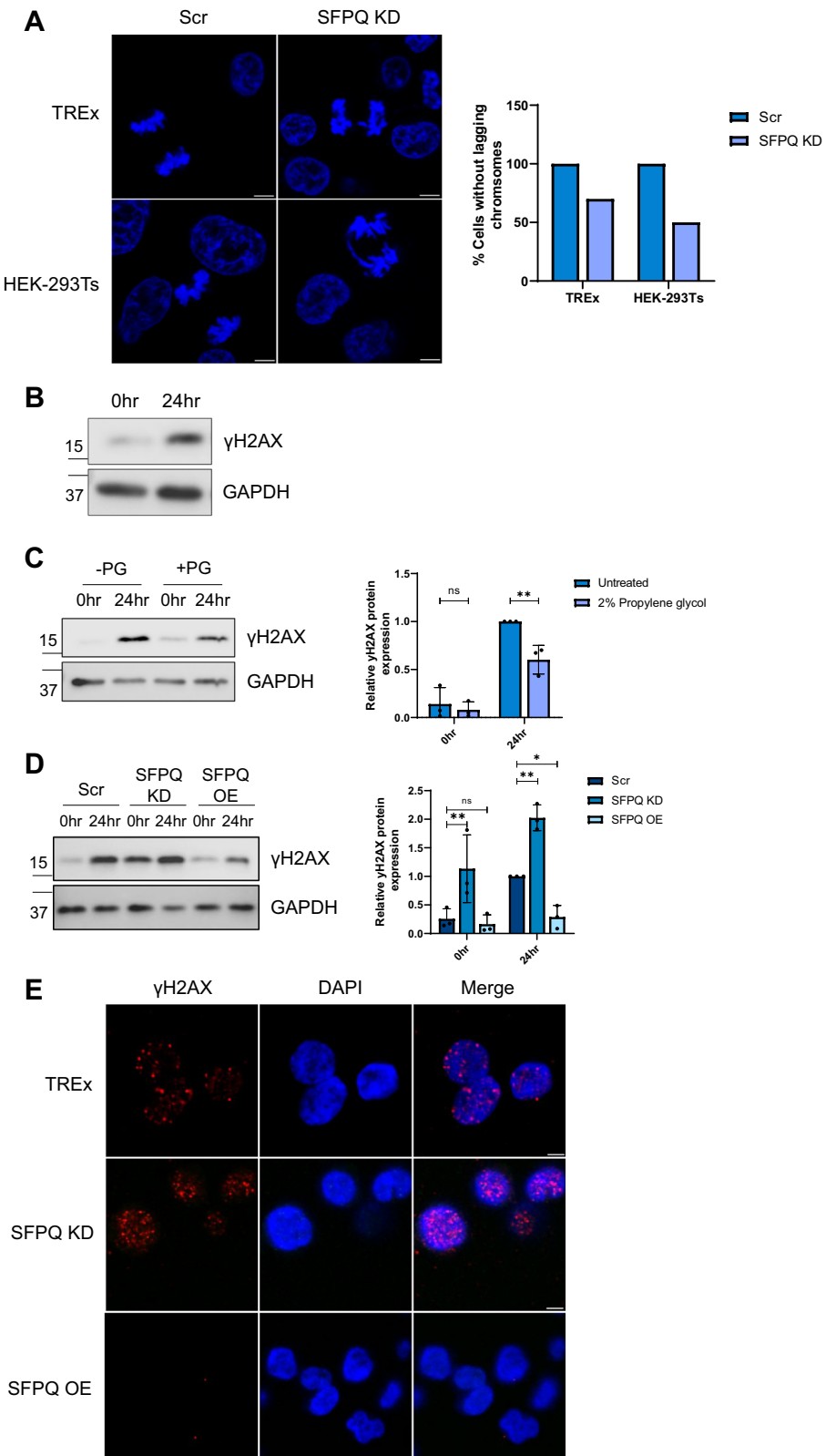

the hypothesis that v-mPS formation, leading to SFPQ sequestration contribute to genomic instability associated with gamma-herpesvirus lytic replication.

Finally, the implications of SFPQ sequestration into v-mPS and increased genomic instability were investigated. It has been previously observed that SFPQ can interact with the Ku70/80 dimer and act as a scaffold for the pre-ligation complex, improving effectiveness of the DDR machinery, notably Ku70 was also enriched within the SFPQ TMT-MS[43,72,73]. Therefore, the interaction between SFPQ and Ku70 was assessed in KSHV latent and lytic TREx-BCBL1-RTA cells. Co-immunoprecipitation assays confirmed an interaction between SFPQ and Ku70 in latent cells, however, this interaction was significantly reduced during KSHV lytic replication (Fig. 10D). Conversely, when v-mPS formation was prevented (using the ORF11 CRISPR cell line) the

**Fig. 9 | Formation of v-mPS disrupts the DDR. A** IF of scr and SFPQ KD latent TREx and HEK-293Ts. Cells were stained for DAPI (blue). 20 cells undergoing active mitosis were counted for each cell type to identify the percentage of lagging chromosomes. **B** Representative western blot for levels of γH2AX in TREx-BCBL1-RTA cells at latent and 24 h post-lytic induction. GAPDH was used as a loading control (n = 3). **C** Representative western blot for levels of γH2AX in TREx-BCBL1-RTA cells at latent and 24 h post-lytic induction. Cells were treated with propylene glycol (PG) between 16 and 24 h post-reactivation. GAPDH was used as a loading control. Densitometry analysis was performed on n = 3. *P* values are 0.6078 and 0.0098. **D** Representative western blot of γH2AX in scr, SFPQ KD and SFPQ OE TREx-BCBL1-RTA at latent and 24 h post-lytic induction. GAPDH was used as a loading control, densitometry analysis was performed on n = 3. *P* values are 0.0055, 0.8932, 0.0018 and 0.0197. **E** IF of γH2AX (red) and DAPI (blue) in latent scr, SFPQ KD and SFPQ OE TREx-BCBL1-RTA. Scale bars are 5 μm. In (**C**, **D**) data are presented as mean ± SD. *\*P* < 0.05, *\*\*P* < 0.01 and *\*\*\*P* < 0.001 (unpaired two-tailed Student's *t* test for (**C**) and ANOVA for (**D**). All '*n*' repeats are biological in Fig. 9. Source data are provided as a source data file.

interaction between SFPQ and Ku70 is increased compared to the scramble control (Fig. 10E). Together, these data suggest that formation of v-mPS leads to an increase in dsDNA breaks due to SFPQ sequestration, which reduces the interaction with Ku70, which in turn may contribute to gammaherpesvirus oncogenesis.

In summary, these findings identify SFPQ and the KSHV-encoded protein ORF11 as key drivers in the formation of novel v-mPS which may act as hubs for RNA processing during KSHV lytic replication (Fig. 10F). Formation of these novel v-mPS also promote genomic instability during lytic replication due to SFPQ sequestration, by preventing its interaction with Ku70 and potentially affecting recruitment of the pre-ligation complex in the DDR. This highlights a novel mechanism which increases DNA damage during gammaherpesvirus lytic replication which may be associated with viral-mediated tumourigenesis.

## Discussion

The dynamic nature of biomolecular condensates allows fast changes to cellular stimuli, enabling a fine-tuned cellular response to a range of conditions in a compartmentalised manner. However, they are also frequently dysregulated in disease states such as cancer, Alzheimer's disease and fronto-temporal dementia[77–80]. Wider research has demonstrated viruses are able to either induce or alternatively disrupt condensate formation to enhance their replicative cycles or regulate a host cell response, such as inhibition of stress granules or formation of viral factories[45,81]. This study has identified a novel gammaherpesvirus-induced v-mPS, which may be involved in RNA processing during infection. Critically, the formation of these structures are essential for successful viral lytic replication, whilst simultaneously increasing DNA damage through sequestration of PSP proteins, thus potentially contributing to gammaherpesvirus-mediated tumourigenesis.

The establishment of v-mPS occurs through LLPS, initially forming a dynamic, liquid-like body that matures into a gel-like state over the first 24 h of viral lytic replication. Importantly, although they utilise many canonical paraspeckle components including *NEAT1*, SFPQ and NONO, there are several key differences, marking them as paraspeckle-like. The first major difference is highlighted by high-resolution microscopy which showed v-mPS are up to 10x larger than canonical paraspeckles and, importantly, their shape remains highly circular, which contrasts with the elongation model previously observed in canonical paraspeckles[82].

Secondly, canonical paraspeckle formation occurs co-transcriptionally with *NEAT1* expression correlating with the number or size of paraspeckles. *NEAT1* containing RNPs then form through binding of SFPQ and NONO, with multiple RNPs then being joined through FUS to form a mature paraspeckle. Surprisingly however, qPCR analysis confirmed *NEAT1* levels remain stable throughout KSHV viral replication despite the increase in paraspeckle-like number and size. Moreover, a combination of TMT-MS analysis, immuno-fluorescence and co-immunoprecipitation studies highlighted several of the core PS proteins are missing, most notably FUS, with immuno-fluorescence confirming FUS being partly re-localised to the vRTCs. Notably, loss of core PSPs, such as FUS, usually inhibits canonical paraspeckle formation[83,84], conversely formation of v-mPS appear to be independent of FUS, implicating a distinct architecture and biogenesis pathway that may utilise different factors or mechanisms.

The functionality of this ring-like structure requires further investigation, with v-mPS at earlier time points appearing as solid puncta. Whereas ring-shaped or toroidal v-mPS were observed at later time points, however, whether this ring structure is linked to maturation of the v-mPS is unknown. As canonical paraspeckles exhibit a core and shell structure[6] it could be hypothesised that v-mPS form with an expanded variant of this with an altered set of structural proteins. This maturation into a ring-like structure may facilitate viral replication by conforming more closely to the surface of the vRTC. In turn, this could help increase the available surface area for the trafficking of RNA and proteins between the two structures thus aiding viral replication.

Alternative biogenesis of v-mPS is likely to be mediated through the KSHV-encoded multifunctional protein ORF11, which co-localises with SFPQ throughout the entirety of v-mPS formation and lifetime. Cells lacking ORF11 fail to form v-mPS and importantly, like many proteins that drive phase separation, ORF11 has low complexity terminal regions, with truncation of ORF11 termini inhibiting v-mPS formation. Phosphorylation of the IDRs located at ORF11 termini may also play a role in v-mPS formation as shown through reduced formation post-mutation, however, confirmation that these serines are phosphorylated and the identity of the kinase targeting them is required to fully elucidate their role. It may be the case that KSHV ORF11 could associate directly with *NEAT1*, enhancing PSP binding which drive seeding events to form stable RNPs. Alternatively, it may enhance RNP fusion, acting as a direct FUS replacement or displacing FUS to form mature v-mPS. Our findings herein suggest the latter, due to the observation that FUS is partially relocalised to vRTCs during infection and the fact that ORF11 is sufficient to rescue paraspeckle formation in FUS-depleted HeLa cells. Whilst similar v-mPS form in EBV, the virus does not contain a direct ORF11 homologue and the closest paralogue to ORF11 in all human herpesviruses, the EBV protein LF2, has only ~28% protein blast alignment sequence similarity. Furthermore, the strain H95-8 lacks LF2, and yet v-mPS were still observed during EBV lytic replication, suggesting there may be core differences between how the two gammaherpesviruses drive v-mPS formation and could have evolved divergently. Additionally SFPQ may have several roles within the EBV lifecycle, as recent findings have shown that SFPQ, but not NONO, helps maintain EBV latency by inducing H1 expression and regulating EBV genomic H1 histone occupancy[69].

The study of dysregulation of individual paraspeckle components by viruses is still in its infancy, however, there are increasing links between multiple viruses and PSPs[34]. These roles range from Influenza virus utilising SFPQ to aid in polyadenylation of its transcripts, whilst multiple picornaviruses re-localise SFPQ from the nucleus to the cytoplasm where it binds viral RNA to promote IRES-mediated translation of viral genes[85–87]. In addition, several core PSPs, such as SFPQ, NONO and PSPC1 associate with viral RNAs, including human rhinovirus, Hepatitis delta virus and SARS-CoV-2, promoting stability and enhancing virus replication[88–90]. Alternatively, paraspeckle components have been implicated in the antiviral response. For example, *NEAT1* expression is increased during Hantavirus infection, which promotes transcription of immune genes through sequestering of the transcriptionally repressive SFPQ. Similarly, *NEAT1* is upregulation in HSV-1 infection via STAT-3 dependent mechanism, thus promoting

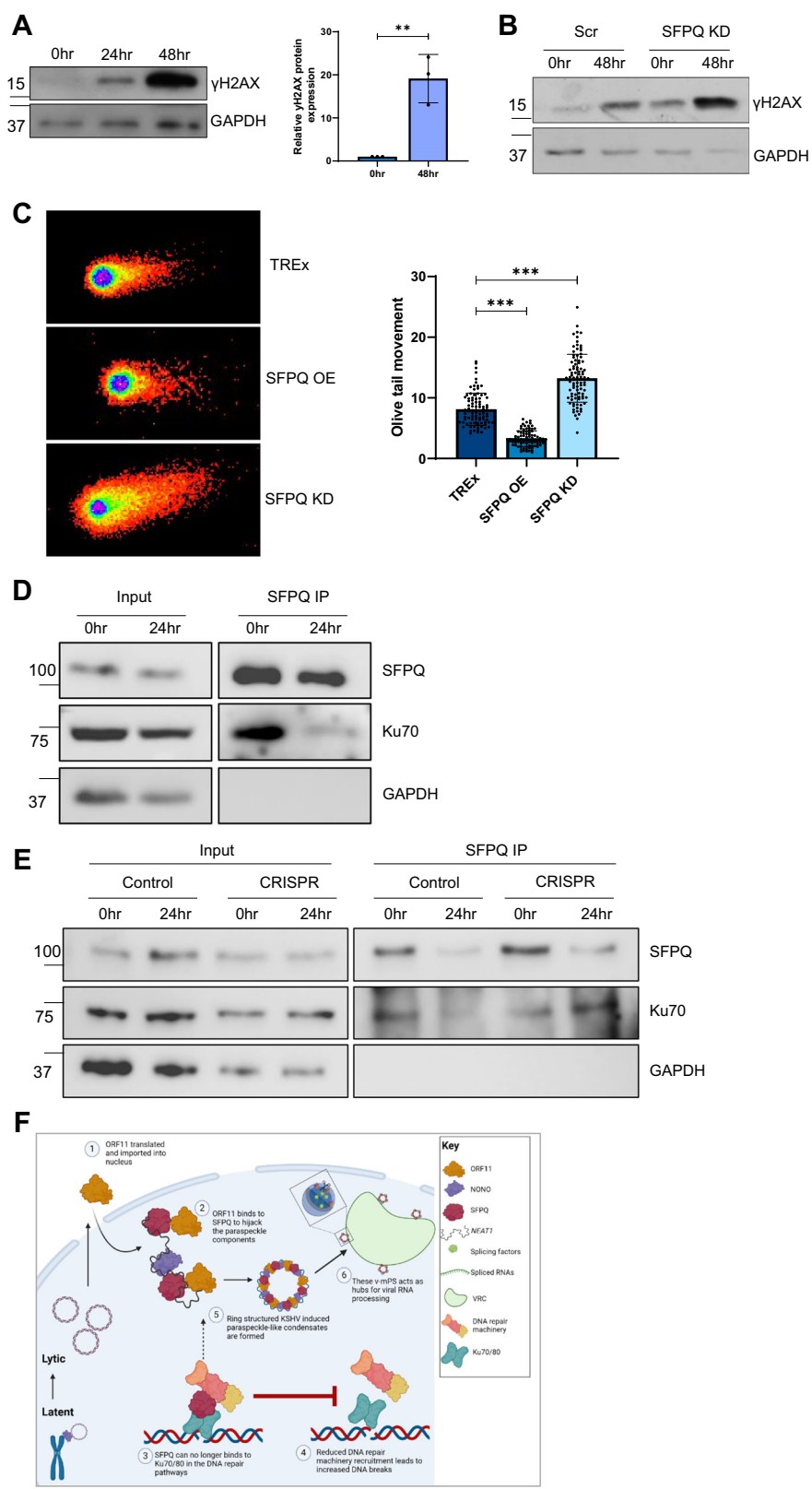

paraspeckle formation[13,91]. Interestingly, PSPC1 has also been shown to regulate HSV-1 gene expression, by enhancing the interaction of STAT-3 with viral gene promoters. Conversely, SFPQ competes with STAT-3 binding to modulate its effects[92]. Therefore paraspeckle components impact viral infection through a variety of mechanisms, however, formation of enlarged v-mPS utilising the vast majority of paraspeckle proteins appears to be gammaherpesvirus specific.

Our results show that v-mPS formation is required for KSHV lytic replication, with depletion of either SFPQ, NONO, *NEAT1*, ORF11 or chemical disruption leading to a significant reduction in the levels of KSHV lytically expressed RNAs and proteins. Our results suggest this reduction is likely due to v-mPS playing a critical role in the processing of viral transcripts, as exemplified by the abundance of viral ORFs identified in the v-mPS by HyPro-Seq and RIPs, that require further

**Fig. 10 | v-mPS formation increases dsDNA breaks. A** Representative western blot for levels of γH2AX in 293-rJJ-L3s cells at latent and 24 and 48 h post-lytic induction. GAPDH was used as a loading control, densitometry analysis for latent and 48 h post-lytic induction was performed on $n = 3$. $P$ value is 0.005. **B** Representative western blot for levels of γH2AX in scr and SFPQ KD 293-rJJ-L3s cells at 0 and 48 h post-lytic induction. **C** Representative neutral comet assays for scr, SFPQ O/E and SFPQ KD TREx-BCBL1-RTA during latency. Olive tail movement was calculated using Comet Score 2.0 ($n = 3$), with 30 cells counted per sample per biological repeat. $P$ values are both <0.0001. **D** Co-immunoprecipitation of SFPQ in TREx cells at latent and 24 h post-lytic induction probed with antibodies against Ku70 and GAPDH ($n = 3$). **E** Co-immunoprecipitation of SFPQ probed with antibodies against Ku70 and GAPDH ($n = 3$) in control and CRISPR ORF11 at latent and 24 h post-lytic

induction. **F** Proposed mechanism for KSHV induction of paraspeckle-like condensates. During KSHV lytic replication, ORF11 is translated and traffics into the nucleus[1]. ORF11 then starts to interact with core paraspeckle proteins, including SFPQ, to drive the formation of the v-mPS[2]. Simultaneously as SFPQ is recruited to the condensates there is an impairment in the DNA damage response, with a reduction in Ku70 binding[3], this leads to an increase in DNA breaks[4]. The fully mature ring-structured v-mPS[5] cluster around the viral replication centres where they act as essential hubs for RNA processing[6]. Image was created in Biorender (Created in BioRender. Whitehouse, (**A**). (2023) https://BioRender.com/x60w950). **A**, **C** Data are presented as mean ± SD. *$P < 0.05$, **$P < 0.01$ and ***$P < 0.001$ (**A**: unpaired two-tailed Student's $t$ test except for (**C**): one way ANOVA). All '$n$' repeats are biological in Fig. 10. Source data are provided as a source data file.

processing and modification, such as *K8*. It is also interesting to note that the levels of both virally-encoded and cellular circRNAs are downregulated upon v-mPS disruption by SFPQ depletion. Although the exact role is still to be elucidated, the enrichment of RNA helicases and hnRNPs in quantitative TMT LC-MS/MS combined with enrichment of these circRNAs in SFPQ-RIP analysis highlights a possible sequestration of these RNA processing factors in circRNA biogenesis. This is supported by recent findings showing that the RNA helicase DDX5 in combination with the m6A reader YTHDC1, both of which are enriched within the TMT LC-MS/MS, have been found to specifically enrich the backsplicing of specific circRNAs[93]. Furthermore, several of these circRNAs have long flanking introns and SFPQ has previously been observed to regulate them[14]. Similarly, hnRNP M has also been shown to positively regulate the biogenesis of circRNAs with long flanking introns[94].

One future direction of this work is to determine the potential RNA targeting process for their association with v-mPS structures, as only a subset of viral transcripts are associated with the v-mPS. Many of these viral transcripts are intron-containing whilst there is also an enrichment of circRNAs with long flanking introns. Previous research has highlighted RNAs enriched in paraspeckles or processed by SFPQ often have these features[6,95], thus this may be a key determining factor. Additionally, the nuclear m6A reader YTHDC1, which is critical for directing methylated transcripts to distinct biological fates, alongside its cofactors hnRNP A2B1 and hnRNP C were identified as a novel SFPQ interactors within the TMT LC-MS/MS. Combining this observation, with the fact that, HyPro-Seq highlighted that 90% of the viral ORFs present in the v-mPS have been shown to be m6A methylated, point to the intriguing suggestion that m6A methylation may play a role in determining which transcripts are associated with v-mPS[96–98]. The proximity of this complex suggests it could be orchestrating the translocation of viral transcripts from vRTCs directly to the v-mPS via targeted methylation.

Due to the specificity of v-mPS formation for oncogenic gammaherpesviruses and the observed increase in chromosomal abnormality upon SFPQ depletion, we further explored a potential link between v-mPS formation and genomic instability. An increase in genomic instability during both KSHV and EBV lytic replication has been previously reported[28,99,100], however mechanisms driving DNA damage are yet to be fully elucidated. Both latent and lytic life cycles are essential for virus-mediated tumourigenesis, and specifically regarding lytic replication, it is proposed that over time, the continuous reactivation in infected individuals either undergoing the full lytic cycle or abortive lytic replication leads to increased DNA damage, thus increasing the risk of cancer development[29].

For example, it has been previously shown that the sequestration of hTREX components to viral mRNAs drives R-loop formation as a contributing factor whilst several DDR proteins including RPA32 and MRE11 have been shown to localise to vRTCs[28,99]. Herein, we identify a novel mechanism which contributes to genomic instability in both KSHV and EBV lytic replications, namely the sequestration of SFPQ and possibly other PSPs, into v-mPS resulting in a concomitant increase in

DNA damage. Therefore, combined with enhanced R-loop formation associated with virus-mediated hTREX sequestration, the proposed sequestration of SFPQ into the v-mPS would lead to a 'dual hit' on DNA damage, with the DDR response potentially also being impaired. SFPQ expression inversely correlates with accumulation of DNA damage, specifically DSBs. SFPQ is a known interactor with the Ku proteins, key players in NHEJ, which is the main repair pathway during DSB[73]. During KSHV lytic replication, this interaction is reduced, likely resulting in reduced efficacy of downstream parts of the repair pathway. Moreover, the sequestration of SFPQ into v-mPS is hypothesised to be especially critical during abortive lytic replication, when the early stages of KSHV lytic replication are triggered, however full lytic replication does not occur. This allows KSHV to re-enter into latency and avoids host cell destruction. v-mPS form during early lytic replication, leading to increased DSBs and DNA damage, which can accumulate during abortive lytic replication and contribute to viral-mediated oncogenesis[23,29,71].

It must be noted that limitations of this study lie in the multi-functional nature of both ORF11 and SFPQ which make it difficult to isolate which roles are causing the observed effects on both KSHV and the host cell. Whilst SFPQ is predominantly associated with paraspeckle biogenesis, it also has roles in both the promotion and suppression of DNA damage, transcriptional regulation and the stress response[42,73,78]. Likewise, ORF11 has recently been characterised as the viral mediator of specialised ribosomes during KSHV lytic replication[48]. Interestingly, cells depleted of ORF11 have similar detrimental effects on viral replication to cells that are depleted of NONO, *NEAT1*, or cells that have v-mPS disrupted via PG. All these conditions led to a reduction of over 75% in expression of the late protein ORF65, but no reduction in early ORF57 protein levels, whereas suppression of SFPQ has a more severe effect on KSHV replication, with reductions in both the early ORF57 and late ORF65 proteins. Changes such as these suggest that the more dramatic loss in viral fitness are likely due to the additional roles that SFPQ carries out in the cell whereas the phenotype displayed by ORF11/NONO/*NEAT1* depletion and PG treatment is representative of the effects from loss of m-vPS formation. These changes in phenotype are important to note, as they must be taken in combination to truly elucidate the function and importance of v-mPS during viral infection.

Additionally, many of the results presented in this paper cannot distinguish between phenotypes due to loss of SFPQ or loss of v-mPS because they are so closely intertwined and there is limited technology available to separate these two entities. Whilst many of the findings can be attributed to v-mPS loss, due to similar phenotypes being observed in conditions where v-mPS are disrupted from depletion of proteins (ORF11, NONO) or chemicals (1,6-HD, PG), it is important to note that the loss of SFPQ function will undoubtedly account for the phenotypes seen as well as the loss of v-mPS. Future research will aim to further separate the functionality of v-mPS from SFPQ itself, however this highlights a need for more LLPS and paraspeckle techniques that can target these membraneless organelles without the removal of SFPQ.

To summarise, gammaherpesviruses manipulate core paraspeckle components to form novel v-mPS, which are associated with RNA processing during KSHV lytic replication. Specifically, KSHV utilises the virally-encoded protein ORF11 as a driver for v-mPS initiation and formation. Interestingly, the sequestration of SFPQ into the v-mPS results in increased DNA damage, which may contribute to gammaherpesvirus oncogenesis. Finally, targeting these v-mPS therapeutically may both target viral lytic replication and additionally reduce the risk of cancer development.

## Methods

### Cell culture

TREx-BCBL1-RTA cells, a gift from Professor JU Jung (University of Southern California), a B-cell lymphoma cell line latently infected KSHV engineered to contain a doxycycline-inducible myc-RTA were cultured in RPMI 1640 with glutamine (Gibco), supplemented with 1% P/S (Gibco), 10% FBS (Gibco) and 100 µg/ml hygromycin B (ThermoFisher). KD cell lines were additionally cultured with 3 µg/ml puromycin (Gibco) or 100 µg/ml zeocin (ThermoFisher). ORF11-CRISPR cells have been previously described (Murphy et al.,[48]). HEK-293T and HeLa cells were purchased from the ATCC and cultured in DMEM (Lonza) and supplemented with 10% FBS and 1% P/S.

HEK-293 T-rKSHV.219 were kindly provided by Dr Jeffery Vieira (University of Washington) and were cultured in DMEM (Lonza), supplemented with 10% FBS, 1% P/S and 3 µg/ml puromycin (Gibco). Human foreskin fibroblasts (HFFs) were a gift from J. Sinclair (University of Cambridge) and cultured in DMEM with 10% FBS and 1% P/S. 293-rJJ-L3 cells, a 293 clone #19 carrying a GFP-negative rJJ-L3 Jijoye EBV BAC, were cultured in RPMI 1640 with glutamine (Gibco), supplemented with 1% P/S (Gibco), 10% FBS (Gibco) and 100 µg/ml hygromycin B (ThermoFisher). 293-SL-HB9-862-L1 cells carry a GFP-negative B95-8 BAC and were cultured in RPMI 1640 with glutamine (Gibco), supplemented with 1% P/S (Gibco), 10% FBS (Gibco) and 100 µg/ml hygromycin B (ThermoFisher). HCMV (Merlin) and HSV-1 virus (SC16) stocks were provided by J. Sinclair and S. Efstathiou (University of Cambridge). All cell lines tested negative for mycoplasma. All cell lines were cultured at 37 °C at 5% $CO_2$.

Virus lytic replication in TREx-BCBL1-RTA cells was induced via addition of 2 µg/ml doxycycline hyclate (Sigma-Aldrich). HEK-293 T-rKSHV.219 cells were induced via addition of 20 ng/ml TPA and 3 mM sodium butyrate. EBV cell lines were trypsinised and induced by transfection of even amounts of pBZLF1, pIE-Rta and pBALF4 plasmids using Lipofectamine 2000 (ThermoFisher) and Opti-MEM (ThermoFisher) during seeding.

100 nM *NEAT1* GapmeRs (antisense complementary oligonucleotides that induce target degradation through RNAse H recruitment) (Qiagen) were added to cells for 24 h prior to reactivation. For 1,6-hexanediol treatment, a 3% v/v treatment of 1,6-hexanediol from a 1 M stock was added to cells for 30 s, whilst recovered cells had media replaced after the 1,6-hexanediol treatment followed by a 30 min incubation at 37 °C. All conditions were washed with PBS followed by fixation. For propylene glycol treatment, cells were passaged in hyperosmotic conditions before addition of 4% of propylene glycol for 8 h.

### Plasmid and antibodies

Antibodies used are listed below: GAPDH (Proteintech 60004-1-Ig, WB 1/5000), ORF57 (Santa Cruz sc-135747, WB 1/1000, IF 1/100), ORF65 (CRB crb2005224, 1/100), SFPQ (Proteintech 15585-1-AP, WB 1/500, IF 1/50, RIP 1/50, IP 1/50), NONO (Proteintech 11058-1-AP, WB 1/1000, IF 1/100), PSPC1 (Proteintech 16714-1-AP, WB 1/3000, IF 1/50) FUS (Proteintech 11570-1-AP, WB 1/5000, IF 1/100), FLAG (Sigma F7425, WB 1/500), GFP (Proteintech 66002-1-Ig, WB 1/5000), Ku70 (Proteintech 10723-1-AP, WB 1/2000), γH2AX (CST, WB 1/1000, IF 1/100), SRSF2 (Novus Bio NB100-1774SS, IF 1/250), RNA pol II (Sigma-Aldrich 05-623,

IF 1/50), SFPQ (Proteintech 67129-1-Ig, IF 1/400, IP 1/400), FLAG (Sigma F1804, IF 1/50), hnRNP U (Proteintech 14599-1-AP, IF 1/20), hnRNP M (Proteintech 26897-1-AP, IF 1/50), DDX17 (Proteintech 19910-1-AP, IF 1/10), DDX21 (Proteintech 66925-1-Ig, IF 1/50), DHX9 (Proteintech 67153-1-Ig, IF 1/50), HA (abcam ab9110, IF 1/50), Ea-D (Santa Cruz sc-58121, WB 1/500, IF 1/50, ICP27 (Santa Cruz sc-58121, IF 1/50), pp72 (Santa Cruz sc-69834, IF 1/50).

pVSV.G and psPAX2 were a gift from Dr Edwin Chen (University of Leeds). PLKO.1 TRC cloning vector was bought from Addgene (gift from David Root; Addgene plasmid #10878). GFP-ORF50 has been described previously[101]. FLAG-ORF11 OE and ORF11 CRISPR plasmids have been described previously[48]. ORF11 truncation mutant was cloned via PCR amplification of truncated form of ORF11 from TREx-BCBL1-RTA cell cDNA and cloned used NEBuilder HIFI DNA assembly kit (NEB) into pLenti-CMV-GFP Zeo plasmid (purchased from addgene #17449). ORF11 serine mutant was bought from Genewiz. GFP-SFPQ OE plasmid was generated via PCR amplification of SFPQ from TREx-BCBL1-RTA cell cDNA and cloned using NEBuilder HIFI DNA assembly kit (NEB) into pLenti-CMV-GFP-puro plasmid (purchased from addgene #17448). pBZLF1, pIE-Rta and pBAL4 were provided by Dr Robert White (ICL).

Various 5'UTRs were cloned into psiCHECK-2 vector (Promega) as described previously[48]. Additionally, the promotor region of ORF57 was cloned into psicheck-2 vector (Promega) via Gibson Assembly cloning.

### Lentivirus-based shRNA KD

Lentiviruses were generated by transfection of HEK-293Ts, as previously described[64]. Briefly, per single 6-well, 4 µl Lipofectamine 2000 (ThermoFisher) was used in combination with 1.2 µg pLKO.1 plasmid expressing the shRNA, 0.65 µg psPAX2 and 0.65 µg pVSV.G. 2 days post-transfection, supernatants were collected and filtered using a 0.45 µm filter and used for transductions on the target cells, in the presence of 8 µg/ml polybrene (Merck Millipore). 3 µg/ml puromycin (Gibco) or 100 µg/ml zeocin (ThermoFisher) was added 48 h post-transduction, before KD analysis via qPCR and western blot if appropriate.

### RNA extraction and qPCR

Total RNA was extracted using Monarch Total RNA Miniprep kit (NEB) as per instructions. RNA (1 µg) was reverse transcribed using LunaScript RT SuperMix Kit (NEB). qPCR was performed utilising appropriate primers, cDNA and GoTaq qPCR MasterMix (Promega) on Rotorgene Q (Qiagen) and analysed via ΔΔ CT against a housekeeping gene as previously described[59]. Primers are listed in a supplementary table.

### Promoter luciferase assay

HEK-293T cells expressing GFP-SFPQ overexpression plasmid were transfected alongside wildtype HEK293T cells with 5'-UTR psiCHECK plasmids, and luciferase activity was measured as described previously[48].

### Immunoblotting

Cell lysates were separated using 8-15% polyacrylamide gels and transferred to Amersham Nitrocellulose Membranes (GE healthcare) via Trans-blot Turbo Transfer system (Bio-Rad). Membranes were blocked in 5% milk in 1xTBS-T or 5% BSA in 1xTBS-T dependent on primary antibody. Membranes were probed with appropriate primary antibodies before secondary IgG HRP conjugated antibodies (Dako Agilent). Proteins were detected with ECL Western Blotting Substrate (Promega) or SuperSignal™ West Femto Maximum Sensitivity Substrate (ThermoFisher) before visualisation with G box (Syngene). Alternatively, Dylight 800 and 600 secondaries (ThermoFisher) were utilised before visualisation with LICOR.

## Immunofluorescence and RNA Fluorescence in situ hybridisation (FISH)

Cells were seeded onto coverslips. For suspension cells, coverslips were pre-treated with poly-L-lysine and cells allowed to settle for 3 h prior to reactivation. Cells were fixed with 4% PFA for 15 min and permeabilised with PBS + 1% triton for 15 min. All further incubation steps occurred at 37 °C. Coverslips were blocked for 1 h in PBS and 1% BSA, followed by 1 h incubation in the appropriate primary antibody and 1 h in Alexa-fluor conjugated secondary antibody (Invitrogen, 1/500)[102]. Coverslips were mounted onto slides using Vectashield Mounting Medium with DAPI (Vector laboratories). Images were obtained using Zeiss LSM880 Inverted Confocal Microscope and processed using Zen Blue Software[101]. For RNA FISH, cells were seeded out and visualised per IF protocols whilst FISH was performed using ViewRNA Cell Plus Assay (ThermoFisher).

## DNA FISH

DNA FISH probes were made using FISH TAG™ DNA Multicolour kit (Invitrogen) according to manufacturer's protocol from ORF50 over-expression plasmid. Cells were seeded onto coverslips and reactivated as per IF protocols. Cells were fixed in 4% PFA for 20 min followed by incubation in 0.1 M HCl for 15 min. Samples were neutralised with 2x saline-sodium citrate (SSC) for 5 min. Coverslips were incubated in equilibrium buffer (15% dextran sulphate v/v, 50% formamide v/v, 1% Tween-20 v/v and 2x SSC) for 1 h. In situ hybridisation was initially performed at 85 °C for 7.5 min before continuing at 37 °C for 18 h. Coverslips were washed in decreasing concentrations of SSC (2x, 1x, 0.1x) at 42 °C for 5 min each. Coverslips were antibody stained using standard IF protocol before nuclear stained with Hoechst for 5 min and mounted with SlowFade Gold antifade reagent.

## Fluorescence recovery after photobleaching (FRAP)

TREx-BCBL1-RTA cells overexpressing GFP-SFPQ were used in FRAP experiments, cells were seeded onto poly-L-lysine-treated glass bottom dishes (Greiner). Bleaching occurred using 488 nm laser at 100% intensity. Fluorescent intensity was measured pre-bleach and every 5 s for 210 s. A non-bleached area was measured concurrently as a control. For each biological repeat, 20 puncta were measured and averaged.

## Stimulated emission depletion microscopy (STED)

TREx-BCBL1-RTA cells were seeded onto washed poly-L-lysine treated cover slips and reactivated 3 h later. Cells were fixed, permeabilised and blocked as per IF. Primary antibodies were used at twice the IF concentration and incubated for an hour at 37 °C. Secondary antibodies were Abberior Star Red (Abberior) and Abberior STAR 580 (Sigma Aldrich) used at 1/100 at 37 °C. Coverslips were mounted in Prolong Gold (ThermoFisher) and visualised on an Axio Observer microscope (Zeiss) with STEDYCON module (Abberior). Deconvolution of STED images was performed using Huygens Software (Scientific Volume Imaging). For the rendering of 3D volumes, STED Z-stacks were acquired as above and deconvolution performed in Huygens Software (Scientific Volume Imaging) before Surface rendering using Imaris Software (Oxford Instruments).

## Protein purification

Wildtype ORF11 cloned into popinJ vector (Addgene: #26045) was transformed into Lemo21(DE3) competent cells and innoculated in 1 L of ampicillin selective LB broth at 37 °C with 180 RPM shaking until $OD_{600}$ reached 0.6 nm absorbance. The culture was induced with IPTG (0.4 mM) and incubated overnight at 18 °C with 180 RPM shaking. Pellets were recovered via centrifugation at 4000$g$ for 20 min at 4 °C, then lysed in lysis/wash buffer (50 mM Tris pH 7.6, 300 mM NaCl, 10 mM imidazole, 5% glycerol) supplemented with 1 mL lysozyme (30 mg/mL) for 20 min on ice. Cell lysate was sonicated on ice for 10 bursts of 10 s on/30 s off at 60% amplitude then lysate was cleared via centrifugation at 25,000$g$ for 40 min at 4 °C. Ni-NTA agarose beads were equilibrated in lysis/wash buffer three times followed by incubation with the cleared lysate for 90 min at 4 °C with end-over-end mixing. Unbound protein was removed by centrifugation at 500$g$ for 5 min, followed by two washes with the lysis/wash buffer. The beads were resuspended in elution buffers (50 mM Tris, 300 mM NaCl, 5% glycerol) with increasing imidazole (80 mM-400 mM) for 15 min with end-over-end mixing followed by centrifugation at 500$g$ for 5 min at 4 °C. All elutions were kept, and aliquots were run on an SDS-PAGE and visualised for purified protein via coomassie blue staining.

## Droplet formation assay

Either purified ORF11-GST or GST protein was gently combined with 10% (w/v) PEG and aliquoted onto to 35 mm glass bottomed dishes (Greiner). Samples were incubated for 2 h at room temperature then visualised for the presence of droplets using LSM800 inverted microscope brightfield with DIC prism attachment on the 40x objective[103].

## FIDA droplet formation assay

20 μM of purified ORF11 was added to FIDABio. Buffer was run through capillary at 3500 mbar for 120 s followed by sample at 2000 m bar for 90 s. All samples were at 25 °C. FIDA was used as manufacturer states, with sample flowing through capillary tubes, fluid velocity differences between capillary centre and outside leading to sample adopting a parabolic confirmation. Fluorescence emitted by protein is measured against time, with droplet formation leading to changes in flow rate and resulting in fluorescent spikes.

## Viral reinfection and viral load assays

TREx-BCBL1-RTA cells were induced for 72 h, DNA was extracted from cells and viral
genomes measured via qPCR whilst the supernatant was added in a 1:1 ratio with naïve HEK-293Ts cell in DMEM. 24 h after supernatant was added, RNA was harvested for analysis via qPCR.

## RNA immunoprecipitations

SFPQ and FLAG RIPs were performed in TREx-BCBL-1 RTA cells using EZ-Magna RIP RNA binding Immunoprecipitation Kit (Merck Millipore) as per manufacturer's instructions. RNA was extracted and purified using TRIzol LS (Invitrogen) as per manufacturer's instructions before analysis via qPCR. Samples were analysed using fold enrichment over percentage inputs.

## Co-immunoprecipitations

SFPQ co-immunopreciptationss were performed in TREx-BCBL-1 RTA or FLAG-ORF11 O/E cells. SFPQ antibody was bound to either Protein A or Protein G magnetic beads (ThermoFisher) dependent on antibody species. Cell lysates were incubated overnight at 4 °C with the beads followed by washes. Co-immunoprecipitationss were analysed alongside inputs via western blotting.

FLAG-Trap co-immunoprecipitations were performed in TREx-BCBL-1 RTA FLAG-ORF11 O/E cells as per the manufacturer's protocol (Chromotek). Briefly, pre-washed FLAG-TRAP agarose beads (25 μL per sample) were combined with cell lysates, followed by a 2 h incubation at 4 °C with rotation. The beads were washed three times, and samples were analysed alongside inputs via western blotting.

## In-vitro pulldowns

SFPQ, NONO and YAP were in-vitro translated via TNT® quick coupled transcription/translation kit (Promega) as per manufacturer's protocol. Either SFPQ, NONO, SFPQ + NONO or YAP were incubated at 4 °C overnight with purified His-tagged ORF11 bound to Ni-NTA beads (ThermoFisher) in binding buffer (50 mM Tris, 0.6% Triton X-100, 0.2% glycerol, 150 mM NaCl, 0.5 mM 2-mecaptoethanol, 2 mM PMSF). The

beads were washed three times with binding buffer and samples were analysed via western blotting.

## Neutral comet assay

Comet assays were performed as published[104] using standard TREx-BCBL1-RTA cells, SFPQ depleted cells and SFPQ OE cells. Cells were suspended in low-melt agarose and added to microscope slides. Slides were incubated overnight at 37 °C in lysis buffer (2% sarkosyl, 0.5 M Na$_2$EDTA, 0.5 mg/ml proteinase K, pH 8.0) followed by electrophoresis (90 mM Tris, 90 mM boric acid, 2 mM Na$_2$EDTA, pH 8.5) and staining in 2.5 µg/ml of propidium iodide before visualisation on Zeiss LSM880 Inverted Confocal Microscope. Comet assays were analysed using CometScore2.0.

## Tandem-mass tagging coupled to liquid chromatography mass spectrometry (TMT LC-MS/MS)

Immunoprecipitation samples for quantitative MS analysis were sent to Bristol proteomics for TMT-LC-MS. Raw data files were obtained from Dr Kate Heesom (University of Bristol proteomics) and two metrics were derived: abundance change and abundance ratio. The former resulted from control values being subtracted from the sample values, whilst the latter resulted from sample values being divided by control values. To find true interactions percentage cut-offs relative to the bait protein were used: ≥5% abundance change, ≥10 abundance ratio. The proteins that met these criteria in both repeats were plotted on STRING to generate interaction networks for each pull-down. STRING website: https://string-db.org/cgi/input?sessionId=bPbwOPnnK8LL&input_page_show_search=on.

## Hybridisation-proximity labelling of *NEAT1*

Hybridisation-proximity labelling of *NEAT1* coupled with next generation sequencing (NGS) was performed as per[57,58]. Briefly, 5×10$^6$ TREx-BCBL1-RTA cells were seeded onto a 100 mm cell culture dish pre-treated with poly-L-lysine and left to settle for 3 h prior to reactivation. After 24 h of lytic replication, cells were fixed with DSP (0.5 mg/mL) for 45 min at room temperature, washed three times with 20 mM Tris in PBS (pH 7.5), then permeabilised with 0.25% Triton in PBS for 15 min at room temperature. Cells were washed twice, followed by the addition of 6 mL of diluted probe mixture (25 nM DIG-labelled *NEAT1* oligonucleotides in buffer (2xSSC buffer, 10% formamide, 10% dextran sulphate). A control condition was included, in which 6 mL of buffer was added to cells with the absence of the probes. The cell culture plates were incubated at 37 °C overnight in a humidified chamber[58].

Samples were washed before being blocked, then incubated in purified HyPro protein (2.7 ng/mL) before washing again. Biotin-phenol and hydrogen peroxide were introduced to the cells for 1 min followed by immediate quenching. Cells were lysed in high-SDS RIPA lysis buffer before scraping into a 1.5 mL microcentrifuge tube for further lysis. Samples were sonicated for 7 cycles of 30 s ON/ 30 s OFF, incubated at 37 °C for 30 min, then incubated in proteinase K. RNA was extracted from the lysates using TRIzol LS (Invitrogen) as per the manufacturers protocol and 5% from each sample were retained for inputs. MyOne streptavidin C1 pulldowns were performed on the remaining 95% of samples and final purified biotinylated RNA was extracted via the TRIzol LS extraction method coupled with the RNA clean and concentrator kit (Zymo Research). Samples were eluted into nuclease-free water and NGS was carried out by Novogene in a paired end format using an Illumina NovaSeq 6000 Sequencing System at ~40 million reads per sample.

## RNA-Seq Analysis

Raw reads were subjected to adaptor trimming and low-quality reads removal using Trimmomatic v0.39[105] with parameters (ILLUMINACLIP:2:30:10 and SLIDINGWINDOW:4:20). Quality filtered, and adapter trimmed reads were aligned to the GQ994935.1 (GenBank) assembly of the human gammaherpesvirus genome using BWA mem v0.7.17[106] with default parameters. Peaks were called against matched background biotinylation control libraries at q < 0.01 using MACS2[107] under the option of -nomodel and -nolambda with -effective genome size and -shift size set to 1.3 ×105 and length of RNA fragments (150 bp), respectively. Peaks were annotated using the prebed function in the RBPInper R package[108] based on GQ994935 annotation (Arias updated) to identify the associated genes. RNA processing status of the identified genes were determined using[24].

## Statistical analysis and reproducibility

Except otherwise stated, graphical data shown represent mean ± standard deviation of mean (SD) using three or more biologically independent experiments. Differences between means were analysed by unpaired two-tailed Student's t-test. One-way Anova test was used for multiple comparisons. For main figure immunofluorescence images, images shown are representative from multiple biological repeats, with each experiment performed independently for at least 3 biological repeat. Statistics was considered significant at $P < 0.05$, with *$P < 0.05$, **$P < 0.01$ and ***$P < 0.001$.

## Reporting summary

Further information on research design is available in the Nature Portfolio Reporting Summary linked to this article.

## Data availability

Source data for TMT-quantitative mass spectrometry have been deposited to the ProteomeXchange Consortium via the PRIDE partner repository with the dataset identifier PXD048834. Source data for Hypro-RNA Sequencing has been deposited to NCBI GEO with the dataset identifier GSE277122. Source data are provided with this paper.

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

## Acknowledgements

The authors thank Professor Jae Jung (UCLA) for the TREx BCBL1-Rta cell line, Dr Jeffery Vieira (University of Washington) for the HEK-293 T-rKSHV.219, Professors John Sinclair and Stacey Efstathiou (University of Cambridge) for HCMV (Merlin) and HSV-1 virus (SC16) stocks, Dr Edwin Chen (University of Westminster) for the psPAX2 and pVSV.G plasmids and Professor Eugene Makeyev and Karen Yap (Kings College, London) for advice and reagents regarding the Hypro-Seq experiments. We would like to thank Dr. Kate Heesom (Proteomics Facility, University of Bristol, UK) for the proteomics technical service and bioinformatics support. We would also like to thank Dr Iain Manfield, Centre for Biomolecular Interactions facility (University of Leeds) and Dr Ruth Hughes and Dr Sally Boxall at University of Leeds Bio-imaging facility. This work was supported in parts by the MRC (MR/X000060/1), White Rose BBSRC Doctoral Training Partnership in Mechanistic Biology (95519935) and University of Leeds Mary & Alice Smith Endowed Research Scholarship.

## Author contributions

Conceptualisation: K.L.H., E.M.H., A.W.; Data curation: K.L.H., E.M.H., C.H., C.A.A., W.W.; Formal Analysis: K.L.H., E.M.H., C.A.A., R.E.W., A.W.); Funding acquisition: A.W.; Investigation: K.L.H., E.M.H., C.H.; Writing—original draft: K.L.H., E.M.H., A.W.; Writing—review & editing: all authors.

## Competing interests

The authors declare no competing interests.
