## [Peer Review File · Nature Communications]

REVIEWER COMMENTS

Reviewer #1 (Remarks to the Author):

The manuscript "Virus-induced paraspeckle-like condensates are essential hubs for gene expression and their formation drives genomic instability" by Adrian Whitehouse's group illustrates paraspeckle components form a ring-like structure to facilitate gammaherpesviruses replication and genomic instability. Specifically, the authors identified KSHV protein ORF11 functions as a driver for the formation of paraspeckle like condensate. Overall, this manuscript reports an interesting and innovative observation. I only have two experimental suggestions and some minor comments:

- 1.To directly investigate relationship between paraspeckle-like structure and KBHV DNA, the authors should carry out IF and RNA/DNA-FISH against paraspeckle markers (SFPQ, NEAT1, PSPC1, NONO) and KBHV genomic DNA to determine whether they can co-localize and form a ring-like structure.
- 2.Please carry out experiments (such as luciferase assay) to determine whether paraspeckle components regulate viral genes transcription.

Minor point:

- 1.Line 138: Please introduce "speckles" such as its relationship with paraspeckles and its marker proteins (SRSF2).
- 2.Figure 1 and 3: Please unify the typeface of Scale bars.
- 3.Figure 1 and other figures (including Supplementary Figures): Why SFPQ can not form a ring-like structure at 24 hours post viral reactivation?
- 4.Figure 2E: Please add p value.
- 5.Figure S3: Please add information about the experimental process in Fig.S3Aiii as to be well understood. In figure S3Diii, ORF57 should be changed to ORF11.
- 6.Figure 5: Correct the figure legends about E and F, and repeat the gel image about Fig. 5F.
- 7.Figure 6: Please add viral plaque assay or viral genomic DNA examination to determine roles of paraspeckle-like condensates in EBV replication.
- 8.Figure 7: In figure 7A, how the authors selected the cells undergoing active mitosis to avoid subjectivity.
- 9.Figure S6A: Please add statistical analysis.
- 10.Lack information about the primer sequence of k8 α and k8 β .
- 11.Discussion: Please try to discuss why paraspeckle-like condensates exhibited a ring-like structure and how the structure facilitated viral replication. Please discuss roles of paraspeckles in other viral replication and genes expression, such as HSV-1.
- 12.Check and correct the typos and grammatical errors throughout the manuscript (e.g., Line 451, Figure 4B).

Reviewer #2 (Remarks to the Author):

Manuscript Nr: NCOMMS-24-16933

Harper et al., "Virus-induced paraspeckle-like condensates are essential hubs for gene expression and

their formation drives genomic instability”

The authors demonstrate that Kaposi sarcoma associated herpesvirus (KSHV) and Epstein Barr virus (EBV) form distinct nuclear SFBQ containing paraspeckles that support lytic replication of these two γ -herpesviruses, but are not found during other herpesvirus replication. These results were observed for lytic KSHV reactivation in both B and epithelial cells. These paraspeckles contained canonical RNAs and proteins but lacked FUS. These non-canonical condensates seemed to be dependent in their formation on KSHV ORF11. Intrinsically disordered regions of ORF11 that drive paraspeckle formation were identified. Silencing of SFBQ, NONO and the RNA NEAT1 compromised lytic protein expression and infectious virus production. The authors further define a role of these virus induced paraspeckles in K8 RNA splicing and both viral and cellular circular RNA formation. Similar paraspeckle formation was observed for EBV, but not HSV and CMV. Finally, SFBQ sequestration into ORF11 induced paraspeckles seems to compromise the double stranded DNA response with increased double stranded DNA lesions. From these data the authors conclude that ORF11 induced paraspeckles support viral DNA splicing thereby supporting virus replication, and at the same time allow for more DNA damage, aiding KSHV and EBV associated tumor formation.

These are very interesting findings but it remains unclear if the paraspeckle associated regulation of RNA splicing or circular RNA formation explains the loss of efficient lytic replication, nor if increased DNA damage in lytically replicating cells contributes to associated tumorigenesis for EBV and KSHV.

Major comments:

1. For EBV lytic reactivation of the type II EBV strain Jijoye was investigated. This is an interesting choice, since type II EBV strains are thought to be less B cell transforming. Does paraspeckle formation also hold true for B95-8 virus lytic reactivation?
2. Since lytic replication of both KSHV and EBV will eventually lead to host cell destruction, the authors should explain in more detail how they envision that enhanced DNA lesions upon SFBQ sequestration into paraspeckles during lytic replication would be involved in KSHV and EBV associated tumor formation. Is there any evidence from tissue staining in KSHV or EBV associated tumors that double-stranded DNA breaks are preferentially observed in cells undergoing lytic replication?
3. The authors describe a role of the virus induced paraspeckle formation in K8 splicing and circular RNA accumulation. Which of these is responsible for the diminished lytic KSHV replication? Would overexpression of K8 splice forms rescue KSHV virion release in the absence of paraspeckle formation?

Minor comments:

1. line 291: additional instead of aadditional; line 543: is a “that” lacking after “It is clear”?

Reviewer #3 (Remarks to the Author):

In this paper, Harper et al. describe their discovery of a new gammaherpesvirus-induced nuclear body. They show that the body is similar, but not identical to a paraspeckle, and it is induced at approximately 16-24 hrs after lytic reactivation of the virus from latency. The bodies are adjacent to replication

compartments suggesting an important function in viral gene expression or perhaps host response to virus. The former possibility is supported by the loss of viral replication upon knockdown of factors necessary for the condensates. Additionally, they show that the viral protein ORF11 localizes to the bodies, interacts with condensate component SFPQ in an RNA-dependent fashion, and that ORF11 is necessary but not sufficient for paraspeckle-like body formation.

Generally, this is a very good observational study. The data describing the nuclear bodies are strong and provide an important novel insight into KSHV manipulations of host cell biology. That said, I have three major concerns on the paper listed below. The first two should be addressed before publication in any journal as they point out a central over-interpretation and one possible conflicting piece of data from the same group. The third point, on mechanism and/or virology, needs to be addressed to expand the impact of the paper beyond that of a specialized journal.

Major:

1. Throughout the manuscript, they over-interpret observations from cells in which SFPQ or other factors are knocked down to be due to paraspeckle-like bodies. For example, “Paraspeckle-like condensates are essential for KSHV lytic replication”, or “Paraspeckle-like condensates are involved in both viral and cellular RNA processing”. They do not, however, demonstrate this. Instead, they demonstrate that paraspeckle components are required. To be absolutely clear, I do not expect that they separate condensate formation from SFPQ/et al. activity. This would be a very difficult experiment to design; in fact, I’m not even sure how one would do that. Nonetheless, they must acknowledge this limitation and interpret their experiments correctly. As it stands now these are strictly correlative observations.

2. An even more important concern is that their previous data (Murphy et al., 2023) do not support their hypothesis that it’s the paraspeckle-like particles. The data in the current paper show that both SFPQ knockdown and ORF11 knockout similarly affect paraspeckle-like condensate formation at 24hr (Fig 3E, 3G, 4B). In addition, SFPQ knockdown leads to a dramatic loss of ORF57 RNA and protein expression at this time point (Fig 4C, 4D). However, ORF57 protein levels are virtually unchanged at this same time point (and maybe even a little higher) in the ORF11 knockout cells (Murphy et al. 2023, Figure 4f). Therefore, since both of these treatments have similar effects on paraspeckle-like condensate formation, but only one has effects on expression of an early gene marker, it seems likely that the effect is not due to paraspeckle-like condensate formation.

3. Despite having interesting results with ORF11 and other factors, very little mechanism or virology is presented. They showed some interesting results with ORF11 IDR domains, but don’t really follow up on this at all. As such, it seems like ORF11 might be involved, but I don’t think they have really proven this definitively. Moreover, many of the downstream phenotypes on mRNA processing (Fig 5E, 5F, 5H) are not particularly compelling despite the dramatic change in nuclear bodies at this time point. This further supports the idea that they ascribed functions may not be directly tied to the nuclear body formation.

Minor:

1. More intuitive labels in figures would improve readability. For example, instead of labeling gels with “KD”, label as “SFPQ KD”. Similarly, label the co-IP figures with specific IP antibody and Western blot

antibody, and show what is tagged. Ideally, one could be able to understand the experiment without reading the legend.

Reviewer #4 (Remarks to the Author):

The study by Harper et al. examines paraspeckles structures and their components during KSHV infection. SFPQ and ORF11 (of the virus) are required for the formation of these structures, that change in size and in composition during infection. The authors connect these structures to viral replication, RNA processing and DNA damage, suggesting that these nuclear bodies contribute to genomic instability during lytic KSHV infection.

The study encompasses many experiments and approaches, while imaging of these proteins and bodies is a major part of the study. One of the main issues with the presentation of the data is the quality of the images. A) It is hard to see the staining in many of the images, they are very dark. In particular, the blue DAPI on black background is hard to see many times. In some images the staining presented is very weak and then in some it is very strong, even saturated perhaps. Also, one gets the feeling that there are antibody penetration issues in some of the images (perhaps ineffective membrane permeabilization or not enough time with antibody incubation), making one wonder about the quality of the antibody/ies used. In contrast, other figures show strong diffuse signal, which is altogether different. Also, zoom ins would have helped in some of the images. B) The whole way the figures and plots are laid out and presented is not very common. C) The initial assumption there are no paraspeckles before infection is strange since one does see many foci, and perhaps the staining of NEAT1 should have been used throughout instead of or in addition to SFPQ.

In general, the second part of the paper and the effects on the cell biology of the cells during infection is better presented and more convincing than the first part, in which the first 2 figures give the impression of over interpreted data and poorly performed experiments.

* Line 132 – “in KSHV-latently infected TReX-BCBL1-RTA cells, no paraspeckles were observed and SFPQ remained diffuse throughout the nucleus” – actually there are nuclear foci in the staining of the 0 hr timepoint – aren't these paraspeckles? (Fig. 1Ai – add arrows to point to paraspeckles). This is also seen in many other images in the paper.

* Line 136-137- "In contrast, during early stages of KSHV lytic replication SFPQ coalesced into distinct large puncta". Can the change be measured?

* Fig S1-Bii - SFPQ seems to stain the nuclear envelope. Is this true?; then in Fig. 2A it looks nucleoplasmic and in 2B its again in the periphery. Seems the same for PSpC1 in 2B, and so it seems that the immunofluorescence was problematic in these experiments and not fully staining the nucleus. In this respect, the staining in paraspeckles sometimes distinctly fills the whole structure, and sometimes has the peripheral shell-like pattern. How do we know this is not some kind of staining Ab penetration issue? See also Fig. 2B, 2K.

* DAPI staining in Fig. S2C looks different, condensation wise, to what is presented in figures 1 & 2. Should try to keep to the same stage in order to make comparisons.

* In Fig. 1 it seems that after 24 hrs there are twice the amount of SFPQ foci as compared to the 24 hrs

infected cells in fig. 2. Why is that?

- * Fig. S1B: the lane of the input blot looks exactly the same blot as the lane of IP in the NONO row!
- * Fig. 2C – we don't really know that these cells have FUS knockdown in them.
- * Lines 168-170: Why would a co-IP show such robust interaction between FUS and SFPQ (and a decrease after infection when FUS is sequestered in RTCs), if at 0hr infection they do not appear colocalized in the IF images and SFPQ is absent from paraspeckles? (same for NEAT1)
- * Fig. S2A,B – the authors should say something about the changes that might be occurring in the amount of some of the proteins during infection e.g. see the staining of NONO at the two times. Seems like the amount of NONO is dramatically reduced, and the Western blot might be showing this as well. Whereas Fig. 2H-I show otherwise.
- * Fig. S2C: text says that FUS localises to the VRTC, however, in the staining it looks dispersed in the entire nucleoplasm and not only in the VRTCs (as determined from the DAPI).
- * Fig. S2H – hard to understand the scale of the cell/nucleus in both images. Seems different scales altogether. Please show the actual cell or an example of a few.
- * Fig. 2J – what does n=3 mean? If 3 cells or 3 paraspeckles, then this is definitely not enough.
- * Fig. 2K – it is unclear what percentage of the cells in the 24hrs treated timepoint have the foci and those that don't, because both options are seen in the image.
- * Line 194 – “qPCR and immunoblot analysis of SFPQ and NONO also showed no significant increase in mRNA or protein levels during lytic replication. Together these data suggest the puncta are non-canonical paraspeckles, therefore we now refer to them as novel virus-induced paraspeckle-like structures.” How do the authors reach this conclusion? It could be that the paraspeckles just reorganize in lytic infection and get larger. The hypothesis seems unconvincing. If the qPCR results are identical- figs 2 G-I, D-E no increase in NEAT1/NONO/SFPQ levels is seen - it implies that the bodies are reorganizing the existing material that was once diffuse, and that the proportion of NEAT:SFPQ:NONO is the same. This almost implies the opposite, not that these are novel bodies, but rather a reorganization of the same bodies. This is roughly seen in the inverse relationship between number and diameter. 15 at 700 nm and 5 at 1800nm results in almost the same value.
- * Lines 218-220: I believe controls of paraspeckles at 0hr infection is essential. Otherwise, calling the liquid phase separated properties of these bodies post infection distinct and novel, seems premature. How would regular liquid phase separated paraspeckles behave?
- * Mass spec – many of the proteins are actually cytoplasmic proteins – how do the authors see this issue?
- * Fig. 3 – very hard to see the signal in images. The Flag antibody is for ORF11, so write ORF11. It seems that the ORF11 in control cells (0h), has different localization. In the cell membrane (fig. 3D,S3J,G) or in the nucleus (fig. S3G,3D). How would overexpression of ORF11 affect the lytic state? It's hard to determine but in fig. 3D the paraspeckles look smaller. In fig. 3G the full ORF11 looks very different than the "normal" SFPQ paraspeckles.
- * Fig. 3B – Actually in figure 3Bii there is a clear band in the RNase treatment, doesn't this mean it's RNA independent?
- * Figure S3B: what is the difference between the panels? Is each row a different treatment?
- * Fig. S3iii – the two GAPDH blots are identical to the two GAPDH blots in Fig. S3Diii !
- * Fig. S3I – DDX21 actually does not seem to colocalize in most SFPQ foci
- * Line 325 – “and any that did form were smaller and malformed compared to the scrambled control (Fig.S4E).” – the nuclei in the ones that do form seem fine (middle row) and the ones in the bottom row

do not form the foci, and the smaller cells may be after mitosis.

* Fig. 6A: HSV-1 infection only one cell is shown, that doesn't look very healthy. More cells should be shown. Clearly there are some foci in the SFPQ staining. Same for HCMV.

* There is a study that shows that paraspeckles components associate with one of the nuclear speckles components, SRSF2, in HSV-1 infected cells (<https://www.nature.com/articles/s42003-021-02742-6>). So also here staining of NEAT1 is required in HSV-1 and HCMV, as clearly paraspeckles components do redistribute at least in HSV-1 infected cells.

Minor:

* Line 119, 153 - references needed

* Line 131 – “an elliptical configuration” – suggest using more scientific description e.g. concentrated in the nucleoplasmic area in between the condensed chromatin.

* Fig. S2: Would be helpful to have treatment conditions indicated in figure. This goes for cell types as well. Otherwise, figure is very confusing without legend.

* Fig. 3G is not explained well

* Fig. S4A, B -better mark of which graph represents the protein levels and which one the RNA

* Line 327 – GapmeRs – explain

* Line 451 – ref brackets

* Add DAPI to fig descriptions

REVIEWER COMMENTS

We thank the reviewers for their comments and suggestions to improve the manuscript. We provide a point-by-point response below detailing how we have modified the manuscript including additional experiments performed. Changes are highlighted in the marked up version of the revised manuscript.

Reviewer 1:

Main comments:

1: *To directly investigate relationship between paraspeckle-like structure and KBHV DNA, the authors should carry out IF and RNA/DNA-FISH against paraspeckle markers (SFPQ, NEAT1, PSPC1, NONO) and KBHV genomic DNA to determine whether they can co-localize and form a ring-like structure.*

DNA FISH and IF has been performed. Results show that viral DNA localises to the viral replication centres (vRTCs) and is not associated with the virus-modified paraspeckles (Fig.S5A, lines 440-443)

2: *Please carry out experiments (such as luciferase assay) to determine whether paraspeckle components regulate viral genes transcription.*

We have carried out luciferase promoter experiments in SFPQ O/E compared to wildtype 293T cells, using the promoters from a range of KSHV encoded ORFs including both immediate early (ORF57) and late (ORF65). We observed minimal changes in expression of luciferase. This therefore suggests that SFPQ does not directly regulate viral gene transcription (FigS5B, lines 443-448).

Minor points:

1: *Line 138: Please introduce "speckles" such as its relationship with paraspeckles and its marker proteins (SRSF2).*

Text modified line 151-154.

2: *Figure 1 and 3: Please unify the typeface of Scale bars.*

This has been corrected and unified.

3: *Figure 1 and other figures (including Supplementary Figures): Why SFPQ can not form a ring-like structure at 24 hours post viral reactivation?*

We believe that SFPQ is consistently forming a ring-like structure by 24 hours, however in some instances the microscope resolution and zoom limits the capability to observe these rings, dependent on the individual size of the condensates. We now include higher resolution STED images and videos to show SFPQ ring-like distribution in more detail (Fig2Fii, FigS2L). We have also discussed the functionality of the potential ring-like structures in the discussion (lines 654-662).

4: *Figure 2E: Please add p value.*

Added

5. *Figure S3: Please add information about the experimental process in Fig.S3Aiii as to be well understood. In figure S3Diii, ORF57 should be changed to ORF11.*

We have added further details to figure legends and methods (lines 260-264). For S3Diii (now figure S3Jiii), we probed for both FLAG (ORF11) for confirmation of the overexpression and also ORF57 (for confirmation of successful viral reactivation).

Figure 5: Correct the figure legends about E and F, and repeat the gel image about Fig. 5F. The gel image has been changed (FigS5C). Captions have been corrected.

7. *Figure 6: Please add viral plaque assay or viral genomic DNA examination to determine roles of paraspeckle-like condensates in EBV replication.*

Viral genomic DNA has been analysed for EBV JJ scr and SFPQ KD cells and included Results confirm a reduction in viral load (FigS6C, lines 535-538)

8. *Figure 7: In figure 7A, how the authors selected the cells undergoing active mitosis to avoid subjectivity.*

Bias was avoided through randomisation, on the confocal microscopes, the first 20 cells observed undergoing mitosis, with separated chromosomes were counted for each condition, this is now stated in text (line 561).

9. *Figure S6A: Please add statistical analysis.*

Added. *Figure S7Aii*

10. *Lack information about the primer sequence of k8 α and k8 β .*

Added supplementary table 1.

11. *Discussion: Please try to discuss why paraspeckle-like condensates exhibited a ring-like structure and how the structure facilitated viral replication. Please discuss roles of paraspeckles in other viral replication and genes expression, such as HSV-1.*

The ring-like structure and function have been discussed in more detail (Lines 654-662). We have added a paragraph in the discussion, highlighting potential roles of PSPs and paraspeckles in other viruses (lines 684-700).

12. *Check and correct the typos and grammatical errors throughout the manuscript (e.g., Line 451, Figure 4B).*

Corrected, we have also proofread the manuscript to correct other errors.

Reviewer 2:

Main Comments:

1. *For EBV lytic reactivation of the type II EBV strain Jijoye was investigated. This is an interesting choice, since type II EBV strains are thought to be less B cell transforming. Does paraspeckle formation also hold true for B95-8 virus lytic reactivation?*

We have now included IF for B95-8 B cells (HB9s) showing SFPQ/NEAT1 condensates do form upon its reactivation (Fig.S6A-B). We have also discussed the implications of these findings in the discussion (Lines 676-683).

2. Since lytic replication of both KSHV and EBV will eventually lead to host cell destruction, the authors should explain in more detail how they envision that enhanced DNA lesions upon SFBQ sequestration into paraspeckles during lytic replication would be involved in KSHV and EBV associated tumor formation. Is there any evidence from tissue staining in KSHV or EBV associated tumors that double-stranded DNA breaks are preferentially observed in cells undergoing lytic replication?

There is previous evidence of enhanced DNA damage in both KSHV and EBV lytic replication, with dysregulation of many proteins involved in double-stranded DNA breaks which we have discussed in text (line 731-739). Unfortunately we were unable to find tissue sections with staining for EBV/KSHV tumours, especially combined with markers for latent/lytic replication to differentiate reactivated cells or DNA damage markers.

However, we have extended the discussion to highlight the current hypothesis that double-stranded breaks occur during lytic replication and their crucial role for tumourgenesis. Specifically, we discuss the role of abortive lytic replication in EBV and KSHV tumourigenesis (lines 752-757). Here the early parts of the lytic temporal cascade occur, however, it reverts to latency before completion and cell death. As sequestration of SFPQ and therefore enhanced DNA damage occurs early in lytic replication, this would still occur in abortive lytic replication, over time, in the cells undergoing abortive lytic replication, there would be an enrichment of double-stranded breaks.

3. The authors describe a role of the virus induced paraspeckle formation in K8 splicing and circular RNA accumulation. Which of these is responsible for the diminished lytic KSHV replication? Would overexpression of K8 splice forms rescue KSHV virion release in the absence of paraspeckle formation?

In regard to the downstream phenotypes, we highlighted several examples of RNA processing mechanisms (K8, CircRNAs). We believe some of these events do have an impact on KSHV replication, for example CircCDYL knockdown has a dramatic effect on lytic replication, reducing ORF65 levels by 90% (Fig S5Giii). Moreover, we believe the v-mPS have multiple roles in RNA processing events which we speculate have a cumulative effect on viral replication. This is reinforced by additional experiments added to the revised manuscript. Here we have performed HyPro-Seq, using a biotinylated NEAT1 probe to proximity label RNAs within v-mPS. This has identified a large number of viral RNA transcripts (Fig5A) identified within the paraspeckles which we have confirmed via SFPQ and ORF11 RIPs (Fig5C-D). Therefore, we believe these structures process a wide variety of transcripts, even if they are subtle changes when looking at one or two genes, these small changes add up to larger impact on the virus. As such, we don't believe K8 overexpression in isolation would rescue KSHV replication.

Minor points:

1. line 291: additional instead of additional; line 543: is a "that" lacking after "It is clear"?

Corrected.

Reviewer 3:

1. Throughout the manuscript, they over-interpret observations from cells in which SFPQ or other factors are knocked down to be due to paraspeckle-like bodies. For example, “Paraspeckle-like condensates are essential for KSHV lytic replication”, or “Paraspeckle-like condensates are involved in both viral and cellular RNA processing”. They do not, however, demonstrate this. Instead, they demonstrate that paraspeckle components are required. To be absolutely clear, I do not expect that they separate condensate formation from SFPQ/et al. activity. This would be a very difficult experiment to design; in fact, I’m not even sure how one would do that. Nonetheless, they must acknowledge this limitation and interpret their experiments correctly. As it stands now these are strictly correlative observations.

We apologise for the over-interpretation and have changed the wording throughout the manuscript to ‘tone down’ the language, additionally we agree with this reviewer that it will be extremely difficult to distinguish whether effects observed are due to paraspeckle as a whole or individual component. This limitation has now been discussed (lines 774-783). However, NONO depletion, NEAT1 depletion, ORF11 depletion and chemical dissolution (PG) all lead to loss of v-mPS and a similar impact on virus replication.

Moreover, we provide further evidence that viral transcripts are associated with v-mPS using HyPro-Seq. Here a biotinylated NEAT1 probe is used to proximity label RNAs within v-mPS. This has identified a large number of viral RNA transcripts (Fig5A) which we have confirmed via SFPQ and ORF11 RIPs (Fig5C-D).

2. An even more important concern is that their previous data (Murphy et al., 2023) do not support their hypothesis that it’s the paraspeckle-like particles. The data in the current paper show that both SFPQ knockdown and ORF11 knockout similarly affect paraspeckle-like condensate formation at 24hr (Fig 3E, 3G, 4B). In addition, SFPQ knockdown leads to a dramatic loss of ORF57 RNA and protein expression at this time point (Fig 4C, 4D). However, ORF57 protein levels are virtually unchanged at this same time point (and maybe even a little higher) in the ORF11 knockout cells (Murphy et al. 2023, Figure 4f). Therefore, since both of these treatments have similar effects on paraspeckle-like condensate formation, but only one has effects on expression of an early gene marker, it seems likely that the effect is not due to paraspeckle-like condensate formation.

We agree with the reviewer that there are differences in the phenotypes observed between SFPQ and ORF11 depletion studies. We are in agreement that SFPQ has a greater impact on virus replication, reducing both ORF57 and ORF65 expression. We believe this is the result of the multifunctional nature of SFPQ and it may be the case that SFPQ is required earlier in the replication cycle, and as stated above in Reviewer 3, point 1 comment it is extremely hard to distinguish the role of PSPs individually and as a paraspeckle.

However, it must be noted that ORF11 depletion has a similar phenotype to NONO and NEAT1 depletion, and chemical disruption of the v-mPS using PG, with a reduction in ORF65 but not ORF57 (Fig4Iiii-iv, FigS4Gi-ii and FigS7B). With 4 different mechanisms of disrupting

v-mPS formation and leading to the same phenotype we are confident that the v-mPS are important for virus replication.

In addition, as detailed below in Reviewer 3, point 3, specific mutation of ORF11 impact v-mPS formation via IDR truncation and serine mutations. Importantly, rescue studies show these do not affect 57 levels, but have a dramatic loss of ORF65 expression (Fig3Fi-iii). We have discussed the limitations of the paper in depth within the results and discussion (lines 410-417, 758-773).

3. Despite having interesting results with ORF11 and other factors, very little mechanism or virology is presented. They showed some interesting results with ORF11 IDR domains, but don't really follow up on this at all. As such, it seems like ORF11 might be involved, but I don't think they have really proven this definitively. Moreover, many of the downstream phenotypes on mRNA processing (Fig 5E, 5F, 5H) are not particularly compelling despite the dramatic change in nuclear bodies at this time point. This further supports the idea that they ascribed functions may not be directly tied to the nuclear body formation.

We have performed additional experiments to provide more mechanistic insight into the role of ORF11.

1. We have performed several in vitro experiments with purified ORF11 to explore its ability to drive phase separation. Firstly an in vitro droplet formation assay, with purified recombinant ORF11 protein, shows that ORF11 can phase separate and form droplets (Fig 3G, lines 323-328). This was confirmed through Flow Induced Dispersion Analysis (FIDA), which enables accurate quantification of a protein's ability to form droplets (Fig S3H, lines 328-333). Here recombinant protein is injected into a capillary, if the protein is capable of LLPS and forms droplets, the detector will record a signal spike, enabling accurate quantification of a protein's ability to form droplets. FIDA analysis confirmed the ability of ORF11 to form droplets through recording many such signal spikes

2. The ability of ORF11 to phase separate, suggests that ORF11 may have a scaffolding role in v-mPS formation, similar to FUS in canonical paraspeckles. Aligned with the observation that FUS is relocalised from v-mPS to vRTCs during infection, we explored whether ORF11 can act as a FUS replacement. To this end, we assessed whether ORF11 could rescue paraspeckle formation in FUS-depleted HeLa cells, which fail to form paraspeckles. SFPQ and NEAT1 staining, show that overexpression of ORF11 can indeed rescue condensate formation in the absence of FUS (Fig S3li, lines 338-343). It was also interesting to note these rescued paraspeckles are larger than canonical paraspeckles (Fig S3lii, lines 343-346).

3. We have performed additional predictive analysis on the ORF11 IDRs and identified 5 potential serine phosphorylation sites within the IDRs. We then assessed the impact of mutating these serine residues on condensate formation. Here we extended the rescue experiments in Fig. 3E, where we compared the ability of wild type, IDR deletion and the serine mutant constructs to rescue condensates formation in the ORF11 CRISPR cell line. As shown in Fig 3E+Fig.S3F, (lines 307-318) we observe that full length ORF11 rescues condensate formation, in contrast to the IDR mutant construct. Interestingly, the serine

mutant only provides a partial recovery of condensate formation, with some very small puncta formed, thus suggesting a possible role for serine phosphorylation in the formation of condensates. This is further supported by additional experiments assessing whether these ORF11 constructs can rescue virus replication in the ORF11 CRISPR cells. Here only the wild type ORF11 construct can rescue late ORF65 protein expression (Fig3F, lines 318-320). These findings have also been highlighted and discussed in relation to v-mPS formation in the discussion (lines 663-676).

In regard to the downstream phenotypes – please see Reviewer 2, point 3 response.

Minor Points:

1. *More intuitive labels in figures would improve readability. For example, instead of labeling gels with “KD”, label as “SFPQ KD”. Similarly, label the co-IP figures with specific IP antibody and Western blot antibody, and show what is tagged. Ideally, one could be able to understand the experiment without reading the legend.*

We have changed the figure labels to clarify experiments.

Reviewer #4 (Remarks to the Author):

Summary point 1. One of the main issues with the presentation of the data is the quality of the images. A) It is hard to see the staining in many of the images, they are very dark. In particular, the blue DAPI on black background is hard to see many times. In some images the staining presented is very weak and then in some it is very strong, even saturated perhaps. Also, one gets the feeling that there are antibody penetration issues in some of the images (perhaps ineffective membrane permeabilization or not enough time with antibody incubation), making one wonder about the quality of the antibody/ies used. In contrast, other figures show strong diffuse signal, which is altogether different. Also, zoom ins would have helped in some of the images.

We have adjusted many of the IF images. The staining brightness has been adjusted, particularly focusing on the DAPI. We have now provide more zoomed in versions of IF images (e.g. Fig3E). We have retaken images that may have penetration issues, with high concentration of permeabilization reagents. In addition, we have performed further super-resolution STED imaging of v-mPS, showing they have a distinctive ring structure and their association with vRTCs (Fig 2L).

*Summary point B) The whole way the figures and plots are laid out and presented is not very common. In general, the second part of the paper and the effects on the cell biology of the cells during infection is better presented and more convincing than the first part, **in which the first 2 figures give the impression of over interpreted data and poorly performed experiments.***

We have tried to improve the presentation of the figures throughout and the formatting of results is consistent throughout the manuscript. In addition, we have added more analysis for our data and further experiments in the first figures to support the findings (e.g. FigS2A,

D, J). Finally, as stated above reviewer 3, point 1, we have toned down the language to prevent over-interpretation of the findings.

C) The initial assumption there are no paraspeckles before infection is strange since one does see many foci, and perhaps the staining of NEAT1 should have been used throughout instead of or in *addition to SFPQ*.

We have repeated several of the core experiments using NEAT1 staining, in addition to SFPQ staining (Fig2K, S2Kii, S3F), particularly the time course of their formation. It is important to note that although we observe small NEAT1 puncta in latent TReX cells, these puncta fail to colocalise with SFPQ puncta, suggesting these are not true paraspeckles. To confirm this we performed similar staining in HeLa cells, which are used as a model cell line for paraspeckles, and show that NEAT1 and SFPQ colocalise in this cell line (FigS2A, lines 176-183).

** Line 132 – “in KSHV-latently infected TReX-BCBL1-RTA cells, no paraspeckles were observed and SFPQ remained diffuse throughout the nucleus” – actually there are nuclear foci in the staining of the 0 hr timepoint – aren’t these paraspeckles? (Fig. 1Ai – add arrows to point to paraspeckles). This is also seen in many other images in the paper.*

As discussed in the point above, NEAT1 and SFPQ IF has now been repeated at 0hr to show the NEAT1 foci are not co-localising with SFPQ foci, suggesting they are not true paraspeckles. Additionally we have repeated this experiment in HeLa cells (the most common paraspeckle cell line) and shown the NEAT1 foci co-localise with SFPQ, confirming they are paraspeckles (FigS2A).

** Line 136-137- "In contrast, during early stages of KSHV lytic replication SFPQ coalesced into distinct large puncta". Can the change be measured?*

We have quantified the change by counting the number of condensates present per cell (Fig 2D) and average diameter of each condensate (Fig 2E) between 16hr and 24hrs into lytic replication. Although, we cannot compare this to latent as there are few, if any NEAT1/SFPQ colocalised paraspeckles present.

** Fig S1-Bii - SFPQ seems to stain the nuclear envelope. Is this true?; then in Fig. 2A it looks nucleoplasmic and in 2B its again in the periphery. Seems the same for PSPC1 in 2B, and so it seems that the immunofluorescence was problematic in these experiments and not fully staining the nucleus. In this respect, the staining in paraspeckles sometimes distinctly fills the whole structure, and sometimes has the peripheral shell-like pattern. How do we know this is not some kind of staining Ab penetration issue? See also Fig. 2B, 2K.*

IF images have been changed to show other examples of 0hr cells where the staining is more consistent. Additionally, we believe that the v-mPS consistently form a ring-like structure by 24 hours, however in some instances the microscope resolution and zoom limits the capability to observe these rings, dependent on the individual size of the condensates. We now include higher resolution STED images and videos to show SFPQ ring-like distribution in more detail (Fig2Fii, FigS2L). In addition, FRAP experiments utilised a GFP-SFPQ overexpression cell line which formed the same ring like structure, which suggests that this is not an antibody penetration issue (Fig S2M).

* DAPI staining in Fig. S2C looks different, condensation wise, to what is presented in figures 1 & 2. Should try to keep to the same stage in order to make comparisons.

This image is in a different cell line (HEK-293T-rKSHV.219 vs everything else is in TREx). TREx form much more distinct viral replication centres which leads to the distinctive compression of the DAPI observed.

* In Fig. 1 it seems that after 24 hrs there are twice the amount of SFPQ foci as compared to the 24 hrs infected cells in fig. 2. Why is that?

We observe variation in number and size between cell to cell. We believe this is natural variation between cells, the levels of virus infection/how efficient viral replication is occurring and finally the frame of the cell. We have multiple examples of cells showing a few condensates in one frame and then many more in other frames. Overall we have counted these foci across a range of time points to gain an accurate picture (Fig2D, line 208).

* Fig. S1B: the lane of the input blot looks exactly the same blot as the lane of IP in the NONO row!

Apologies, we cropped the wrong side of the blot for the IP (the uncropped version is shown below). We have now changed this figure to a new IP so it didn't have to be a composite blot.

* Fig. 2C – we don't really know that these cells have FUS knockdown in them.

qRT-PCR and western blotting show that FUS knockdown is 95% KD at RNA and 90% at protein levels. We have now included in Fig S2H, IF of scrambled vs FUS depleted cells stained with SFPQ and FUS, showing that FUS is depleted but v-mPS still form. Additionally, we have now used this lentivirus KD in HeLa cells, resulting in similar levels of KD and unlike the TREx FUS knockdowns, FUS knockdown in HeLa does lead to paraspeckle inhibition (FigS2H-J).

* Lines 168-170: Why would a co-IP show such robust interaction between FUS and SFPQ (and a decrease after infection when FUS is sequestered in RTCs), if at 0hr infection they do not appear colocalized in the IF images and SFPQ is absent from paraspeckles? (same for NEAT1)

We believe that although the distinct structural paraspeckles are not forming, the individual components still reside in the nucleoplasm. They may be interacting outside their paraspeckle roles as both classical RNA binding proteins. In addition, these Co-IPs are not formed in the presence of RNase.

* Fig. S2A,B – the authors should say something about the changes that might be occurring in the amount of some of the proteins during infection e.g. see the staining of NONO at the

two times. Seems like the amount of NONO is dramatically reduced, and the Western blot might be showing this as well. Whereas Fig. 2H-I show otherwise.

Host-cell shut off occurs during KSHV lytic replication, however, we believe NONO levels are consistent through the infection as shown in (Fig S2C) and (Fig 2G and I). The blots for IPs in Fig S2C has been repeated and new blots included in this revision.

* Fig. S2C: text says that FUS localises to the VRTC, however, in the staining it looks dispersed in the entire nucleoplasm and not only in the VRTCs (as determined from the DAPI).

We agree FUS localisation is not just present in the vRTCs and only a partial localisation is observed. Text has been changed to reflect it is a partial localisation (lines 191-192).

However, to confirm that FUS is relocalised to the vRTCS and not colocalising with SFPQ, we have performed Zen analysis (S2iii-iv). FUS has a similar Zen profile to RNA pol II, and is also distinct from the compressed DAPI.

* Fig. S2H – hard to understand the scale of the cell/nucleus in both images. Seems different scales altogether. Please show the actual cell or an example of a few.

Scale bars have now been added, showing the images have been taken using the same magnification. Size differences in the condensates are due to them still forming at 16hrs.

* Fig. 2J – what does n=3 mean? If 3 cells or 3 paraspeckles, then this is definitely not enough.

Clarified in methods text. N=3, is for biological repeats, with 20 bleachings for each biological repeat.

* Fig. 2K – it is unclear what percentage of the cells in the 24hrs treated timepoint have the foci and those that don't, because both options are seen in the image.

Utilising IF stained with ORF57 (a marker for viral replication) and SFPQ, we have analysed the number of cells for both 16 and 24 hours that do/do not have the condensates in 1,6HD treated cells, with 20% of condensates been resistant at 16 hours and 74% at 24 hours (Fig.S20, lines 241-244)

* Line 194 – “qPCR and immunoblot analysis of SFPQ and NONO also showed no significant increase in mRNA or protein levels during lytic replication. Together these data suggest the puncta are non-canonical paraspeckles, therefore we now refer to them as novel virus-induced paraspeckle-like structures.” How do the authors reach this conclusion? It could be that the paraspeckles just reorganize in lytic infection and get larger. The hypothesis seems unconvincing. If the qPCR results are identical- figs 2 G-I, D-E no increase in NEAT1/NONO/SFPQ levels is seen - it implies that the bodies are reorganizing the existing material that was once diffuse, and that the proportion of NEAT:SFPQ:NONO is the same. This almost implies the opposite, not that these are novel bodies, but rather a reorganization of the same bodies. This is roughly seen in the inverse relationship between number and diameter. 15 at 700 nm and 5 at 1800nm results in almost the same value.

We agree with the reviewer's point on reorganisation of the components, however, this is not the point we were trying to make. Canonical paraspeckle formation occurs co-

transcriptionally, as such to make more canonical paraspeckles NEAT1 has to be actively transcribed, then SFPQ and NONO binding is required to stabilise NEAT1 whilst it is being transcribed. However, even though we observe enlarged formation of paraspeckles, we do not see any increase in NEAT1 transcription, therefore we believe the formation is not happening co-transcriptionally, suggesting a non-canonical formation.

As stated in reviewer 4 point C, we fail to observe any paraspeckles in latent cells, therefore we believe these structures are virus induced and novel. We agree that at 16 hours reactivation, there is no increase in paraspeckle components and therefore they probably reorganise and accumulate, however this is dependent on ORF11, and this separate biogenesis mechanism which doesn't require active NEAT1 transcription is novel. This is now clarified in text (lines 219-227, 643-653).

** Lines 218-220: I believe controls of paraspeckles at Ohr infection is essential. Otherwise, calling the liquid phase separated properties of these bodies post infection distinct and novel, seems premature. How would regular liquid phase separated paraspeckles behave?*

We have now performed IF/FISH on canonical paraspeckles in HeLa cells as well as on the Ohr latent TReX (Fig.S2A) 1,6 HD treatment has been performed on canonical paraspeckles in HeLas which, as per, the literature, dissolved, showing liquid like properties (Fig.S2P, lines 249-251).

** Mass spec – many of the proteins are actually cytoplasmic proteins – how do the authors see this issue?*

We have analysed the TMT-MS results further using STRING analysis to highlight proteins that can localise to the nucleus. Nuclear proteins in the SFPQ IPs are shown in green, Nuclear proteins in the ORF11 Ips are shown in red. From this analysis the vast majority can reside in the nucleus.

** Fig. 3 – very hard to see the signal in images. The Flag antibody is for ORF11, so write ORF11. It seems that the ORF11 in control cells (0h), has different localization. In the cell membrane (fig. 3D,S3J,G) or in the nucleus (fig. S3G,3D). How would overexpression of ORF11 affect the lytic state? It's hard to determine but in fig. 3D the paraspeckles look smaller. In fig. 3G the full ORF11 looks very different then the "normal" SFPQ paraspeckles.*

The signal has been brightened across the images. We have also changed the image in 3G to better represent the rescue and demonstrate the paraspeckles can be rescued to 'normal'. Figure captions been amended. For ORF11 localisation during latency, we consistently see this membrane like localisation, it is worth noting that in normal TReX cells, ORF11 is not expressed at all during latency and this is an artefact of the OE construct we have to use, due to the lack of any antibody reagents. We believe, another viral factor or cellular process that is manipulated during lytic replication leads to the correct localisation for ORF11 during lytic replication.

** Fig. 3B – Actually in figure 3Bii there is a clear band in the RNase treatment, doesn't this mean it's RNA independent?*

Whilst there is still a band in the RNase treated sample, it is reduced compared to the untreated sample. This suggests that RNA interactions may be enhanced, presumably by NEAT1, even if it is not entirely dependent on it. This is supported by pulldown assays using recombinant ORF11 and ITT SFPQ and NONO., which showed a weak protein-protein interaction. Wording has been changed in the manuscript to reflect this (lines 277-282).

** Figure S3B: what is the difference between the panels? Is each row a different treatment? S3B is taken from a single Z stack experiment, with each row a different plane of the cell. This has now been clarified in text and in figure (Fig.S3D, Lines 287.)*

** Fig. S3iii – the two GAPDH blots are identical to the two GAPDH blots in Fig. S3Diii ! Apologies, the wrong blot was included, this has now been changed.*

** Fig. S3I – DDX21 actually does not seem to co-localise in most SFPQ foci*

Arrows have been added to highlight co-localisation with the condensates where it is not as clear. For DDX21, we agree the majority of DDX21 shows a distinct different localisation, which through contrast can make it harder to spot localising to the condensates, however, with improved brightness it can be observed. We have expanded the text, clarifying that the factors co-localising with SFPQ are also observed elsewhere in the cell (lines 364-367).

**Line 325 – “and any that did form were smaller and malformed compared to the scrambled control (Fig.S4E).” – the nuclei in the ones that do form seem fine (middle row) and the ones in the bottom row do not form the foci, and the smaller cells may be after mitosis.*

We have repeated this experiment and utilised RNA FISH to better confirm the result. NEAT1 FISH found the vast majority of reactivated cells (95%+) do not form condensates. Cells where SFPQ showed a redistribution, did not form mature condensates, with no rings observed. Additionally, the puncta were far more characteristic of the phenotype observed pre-16hrs, suggesting NONO depletion does prevent condensate formation.

** Fig. 6A: HSV-1 infection only one cell is shown, that doesn't look very healthy. More cells should be shown. Clearly there are some foci in the SFPQ staining. Same for HCMV.*

** There is a study that shows that paraspeckles components associate with one of the nuclear speckles components, SRSF2, in HSV-1 infected cells*

<https://www.nature.com/articles/s42003-021-02742-6>). So also here staining of NEAT1 is required in HSV-1 and HCMV, as clearly paraspeckles components do redistribute at least in HSV-1 infected cells.

Staining has been repeated for HSV-1 and HCMV in HFFs. We have also repeated the staining, utilising RNA FISH against NEAT1 (Fig6B). We do not observe the formation of the large condensate structures we observed during EBV/KSHV infection. For both HCMV and HSV-1, there is no significant change in SFPQ staining. For NEAT1 staining, HCMV infection leads to a more diffuse NEAT1 staining, with a reduction in paraspeckles compared to uninfected. For HSV-1 staining, there was a small increase in the number of NEAT1 puncta/paraspeckles, however, they appear to be canonical, and whether it is a viral response or a cell stress response is unclear. We use HFFs for our infections, with published work utilising HeLa cell lines, with cell type variation possible, particularly as HeLas are known to form paraspeckles under cellular stress. We discuss the staining patterns in more detail (lines 502-510).

Minor points:

Line 119, 153 references needed

Added

* Line 131 – “an elliptical configuration” – suggest using more scientific description e.g. concentrated in the nucleoplasmic area in between the condensed chromatin

Text has been modified (line 142)

* Fig. S2: Would be helpful to have treatment conditions indicated in figure. This goes for cell types as well. Otherwise, figure is very confusing without legend.

Text added.

* Fig. 3G is not explained well

The legend has been modified (now Fig3I) and expanded in main text (lines 361-363)

* Fig. S4A, B -better mark of which graph represents the protein levels and which one the RNA

Changed within axis

* Line 327 – GapmeRs – explain

Added

* Line 451 – ref brackets

Fixed

* Add DAPI to fig descriptions

Added

REVIEWERS' COMMENTS

Reviewer #1 (Remarks to the Author):

I appreciate the authors' efforts in addressing my comments. The study looks promising and the current version of manuscript can be accepted for publication.

Reviewer #2 (Remarks to the Author):

Manuscript Nr: NCOMMS-24-16933A

Harper et al., "Virus-induced paraspeckle-like condensates are essential hubs for gene expression and their formation drives genomic instability"

The authors demonstrate that Kaposi sarcoma associated herpesvirus (KSHV) and Epstein Barr virus (EBV) form distinct nuclear SFBQ containing paraspeckles that support lytic replication of these two γ -herpesviruses, but are not found during other herpesvirus replication. These results were observed for lytic KSHV reactivation in both B and epithelial cells. These paraspeckles contained canonical RNAs and proteins but lacked FUS. These non-canonical condensates seemed to be dependent in their formation on KSHV ORF11. Intrinsically disordered regions of ORF11 that drive paraspeckle formation were identified. Silencing of SFBQ, NONO and the RNA NEAT1 compromised lytic protein expression and infectious virus production. The authors further define a role of these virus induced paraspeckles in K8 RNA splicing and both viral and cellular circular RNA formation. Similar paraspeckle formation was observed for EBV, but not HSV and CMV. Finally, SFBQ sequestration into ORF11 induced paraspeckles seems to compromise the double stranded DNA response with increased double stranded DNA lesions. From these data the authors conclude that ORF11 induced paraspeckles support viral DNA splicing thereby supporting virus replication, and at the same time allow for more DNA damage, aiding KSHV and EBV associated tumor formation.

In their revised manuscript the authors have addressed some of my concerns, namely extend SFBQ associated paraspeckle formation to a type I EBV strain. However, they have not managed to obtain tissue sections of double-infected tumor to localize DNA double-strand breaks to lytically virus replicating cells. Instead, they have discussed that this might occur during early abortive lytic replication and then still contribute to genetic lesions in the respective tumors, after these cells revert to latent infection. Finally, they have further broadened content and possible functions of the SFBQ containing paraspeckles but not demonstrated that any of the described functions is responsible for enhanced lytic replication. Therefore, the manuscript is somewhat improved.

Reviewer #3 (Remarks to the Author):

The authors have addressed most of my concerns. I would point out that they do not seem to

acknowledge one of the main thrusts of my point clearly enough in the revised manuscript. Deletion of several factors NONO, SFPQ, NEAT1, ORF11 all show similar reductions in v-mPS formation. However, only SFPQ has a clear effect on ORF57 levels. The authors suggest this is due to an additional function of SFPQ. I agree that that's a reasonable assertion.

However, what I think remains a bit buried in the current manuscript is that, since v-mPS can be lost upon NONO/NEAT1 depletion, and yet ORF57 persists, this is a clear falsification of the hypothesis that v-mPS are necessary for ORF57 expression. Of course, v-mPS do not need to function from early to late stage to be of interest. However, clear interpretations of their data should be given.

Moreover, the data showing that circRNAs are affected by SFPQ knockdown become inconclusive (Fig 5I-L) because SFPQ functions in viral gene expression outside of its role in v-mPS. These data should be removed or tested with NONO kd (as was done in Fig 5G/5H for K8).

Reviewer #4 (Remarks to the Author):

The manuscript is much improved and the image quality is better.

Remarks:

The fact that figures are presented on several pages is uncommon. Also, all the subdivisions of the panels to A,B,C... (and reaching even the letters M,N,O) and the further Bi, Bii, Biii subdivisions are also uncommon and actually makes it hard to follow the text. I would suggest dividing some figures into two figures and using conventional numbering.

Fig. 2D – no control (time 0 or untreated)?

Would suggest making the scale bars more evident.

Western 3Bii is not the best blot...

REVIEWERS' COMMENTS

Reviewer #1 (Remarks to the Author):

I appreciate the authors' efforts in addressing my comments. The study looks promising and the current version of manuscript can be accepted for publication.

No comments to address.

Reviewer #2 (Remarks to the Author):

Manuscript Nr: NCOMMS-24-16933A

Harper et al., "Virus-induced paraspeckle-like condensates are essential hubs for gene expression and their formation drives genomic instability"

The authors demonstrate that Kaposi sarcoma associated herpesvirus (KSHV) and Epstein Barr virus (EBV) form distinct nuclear SFBQ containing paraspeckles that support lytic replication of these two γ -herpesviruses, but are not found during other herpesvirus replication. These results were observed for lytic KSHV reactivation in both B and epithelial cells. These paraspeckles contained canonical RNAs and proteins but lacked FUS. These non-canonical condensates seemed to be dependent in their formation on KSHV ORF11. Intrinsically disordered regions of ORF11 that drive paraspeckle formation were identified. Silencing of SFBQ, NONO and the RNA NEAT1 compromised lytic protein expression and infectious virus production. The authors further define a role of these virus induced paraspeckles in K8 RNA splicing and both viral and cellular circular RNA formation. Similar paraspeckle formation was observed for EBV, but not HSV and CMV. Finally, SFBQ sequestration into ORF11 induced paraspeckles seems to compromise the double stranded DNA response with increased double stranded DNA lesions. From these data the authors conclude that ORF11 induced paraspeckles support viral DNA splicing thereby supporting virus replication, and at the same time allow for more DNA damage, aiding KSHV and EBV associated tumor formation.

In their revised manuscript the authors have addressed some of my concerns, namely extend SFBQ associated paraspeckle formation to a type I EBV strain. However, they have not managed to obtain tissue sections of double-infected tumor to localize DNA double-strand breaks to lytically virus replicating cells. Instead, they have discussed that this might occur during early abortive lytic replication and then still contribute to genetic lesions in the respective tumors, after these cells revert to latent infection. Finally, they have further broadened content and possible functions of the SFBQ containing paraspeckles but not demonstrated that any of the described functions is responsible for enhanced lytic replication. Therefore, the manuscript is somewhat improved.

No comments to address.

Reviewer #3 (Remarks to the Author):

The authors have addressed most of my concerns. I would point out that they do not seem to acknowledge one of the main thrusts of my point clearly enough in the revised manuscript. Deletion of several factors NONO, SFPQ, NEAT1, ORF11 all show similar reductions in v-mPS formation. However, only SFPQ has a clear effect on ORF57 levels. The authors suggest this is due to an

additional function of SFPQ. I agree that that's a reasonable assertion.

However, what I think remains a bit buried in the current manuscript is that, since v-mPS can be lost upon NONO/NEAT1 depletion, and yet ORF57 persists, this is a clear falsification of the hypothesis that v-mPS are necessary for ORF57 expression. Of course, v-mPS do not need to function from early to late stage to be of interest. However, clear interpretations of their data should be given.

Throughout the paper we have discussed the limitation of distinguishing between v-mPS and individual protein effects, in particular a lengthy section in the discussion which highlights that v-mPS are required for the complete viral lytic replication (lines 757-782). We specifically highlight that ORF57 expression is only lost with SFPQ depletion, in contrast NONO, NEAT1, ORF11 depletion and chemical dissolution only affect ORF65 late gene expression. We further highlight that potential role of SFPQ outside v-mPS function.

Moreover, the data showing that circRNAs are affected by SFPQ knockdown become inconclusive (Fig 5I-L) because SFPQ functions in viral gene expression outside of its role in v-mPS. These data should be removed or tested with NONO KD (as was done in Fig 5G/5H for K8).

We have now tested the circRNAs with the NONO KD and included this data (Fig.S5J-K).

Reviewer #4 (Remarks to the Author):

The manuscript is much improved and the image quality is better.

Remarks:

The fact that figures are presented on several pages is uncommon. Also, all the subdivisions of the panels to A,B,C... (and reaching even the letters M,N,O) and the further Bi, Bii, Biii subdivisions are also uncommon and actually makes it hard to follow the text. I would suggest dividing some figures into two figures and using conventional numbering.

Fig. 2D – no control (time 0 or untreated)?

Would suggest making the scale bars more evident.

Western 3Bii is not the best blot...

Figures are now included one page, and we have removed the subsection captions. Figure 2D cannot have a time 0 included as there are no v-mPS to measure. Scale bars have been thickened. The western blot has been changed.